# Risk of long COVID and associated symptoms after acute SARS-COV-2 infection in ethnic minorities: A nationwide register-linked cohort study in Denmark

**George Frederick Mkoma**[ID][1]*, **Charles Agyemang**[2,3], **Thomas Benfield**[4,5], **Mikael Rostila**[6,7], **Agneta Cederström**[6,7], **Jørgen Holm Petersen**[ID][8], **Marie Norredam**[1,4]

1 Danish Research Centre for Migration, Ethnicity and Health, Section of Health Services Research, Department of Public Health, University of Copenhagen, Copenhagen, Denmark, 2 Department of Public and Occupational Health, Amsterdam Public Health Research Institute, Amsterdam UMC, University of Amsterdam, Amsterdam, the Netherlands, 3 Division of Endocrinology, Diabetes, and Metabolism, Department of Medicine, Johns Hopkins University, Baltimore, Maryland, United States of America, 4 Department of Infectious Diseases, Copenhagen University Hospital - Amager and Hvidovre, Hvidovre, Denmark, 5 Department of Clinical Medicine, Faculty of Health and Medical Sciences, University of Copenhagen, Copenhagen, Denmark, 6 Department of Public Health Sciences, Stockholm University, Stockholm, Sweden, 7 Centre for Health Equity Studies (CHESS), Stockholm University/Karolinska Institutet, Stockholm, Sweden, 8 Section of Biostatistics, Department of Public Health, University of Copenhagen, Copenhagen, Denmark

* george.mkoma@sund.ku.dk

**Data Availability Statement:** Data that supports the findings of this work are stored at Statistics Denmark and are not publicly available due to

## Abstract

### Background

Ethnic minorities living in high-income countries have been disproportionately affected by Coronavirus Disease 2019 (COVID-19) in terms of infection rates, hospitalisations, and deaths; however, less is known about long COVID in these populations. Our aim was to examine the risk of long COVID and associated symptoms among ethnic minorities.

### Methods and findings

We used nationwide register-based cohort data on individuals diagnosed with COVID-19 aged ≥18 years (n = 2,287,175) between January 2020 and August 2022 in Denmark. We calculated the risk of long COVID diagnosis and long COVID symptoms among ethnic minorities compared with native Danes using multivariable Cox proportional hazard regression and logistic regression, respectively.

Among individuals who were first time diagnosed with COVID-19 during the study period, 39,876 (1.7%) were hospitalised and 2,247,299 (98.3%) were nonhospitalised individuals. Of the diagnosed COVID-19 cases, 1,952,021 (85.3%) were native Danes and 335,154 (14.7%) were ethnic minorities. After adjustment for age, sex, civil status, education, family income, and Charlson comorbidity index, ethnic minorities from North Africa (adjusted hazard ratio [aHR] 1.41, 95% confidence interval [CI] [1.12,1.79], p = 0.003), Middle East (aHR 1.38, 95% CI [1.24,1.55], p < 0.001), Eastern Europe (aHR 1.35, 95% CI [1.22,1.49], p <

Danish Data Protection Act. Data access may be granted upon approval from the relevant data custodians. More details about data and conditions for access can be found on Statistics Denmark website via https://www.dst.dk/en/TilSalg/Forskningsservice.

**Funding:** This work was supported by the grant from the Novo Nordisk Foundation (https://novonordiskfonden.dk/en/) with the grant number 0067528 for MN. The funders had no role in study design, data collection and analysis, decision to publish, or preparation of the manuscript.

**Competing interests:** TB reports grants from Novo Nordisk Foundation, Lundbeck Foundation, Simonsen Foundation, GSK, Pfizer, Gilead, Kai Hansen Foundation and Erik and Susanna Olesen's Charitable Fund; personal fees from GSK, Pfizer, Bavarian Nordic, Boehringer Ingelheim, Gilead, MSD, Pentabase ApS, Becton Dickinson, Janssen and Astra Zeneca; outside the submitted work. Other authors have declared that no competing interests exist.

**Abbreviations:** aHR, adjusted hazard ratio; CCI, Charlson comorbidity index; CI, confidence interval; COVID-19, Coronavirus Disease 2019; DNPR, Danish National Patient Registry; HR, hazard ratio; NICE, National Institute for Health and Care Excellence; OR, odds ratio; PCR, polymerase chain reaction; SARS-CoV-2, Severe Acute Respiratory Syndrome Coronavirus 2.

0.001), and Asia (aHR 1.23, 95% CI [1.09,1.40], $p = 0.001$) had significantly greater risk of long COVID diagnosis than native Danes. In the analysis by largest countries of origin, the greater risks of long COVID diagnosis were found in people of Iraqi origin (aHR 1.56, 95% CI [1.30,1.88], $p < 0.001$), people of Turkish origin (aHR 1.42, 95% CI [1.24,1.63], $p < 0.001$), and people of Somali origin (aHR 1.42, 95% CI [1.07,1.91], $p = 0.016$). A significant factor associated with an increased risk of long COVID diagnosis was COVID-19 hospitalisation. The risk of long COVID diagnosis among ethnic minorities was more pronounced between January 2020 and June 2021. Furthermore, the odds of reporting cardiopulmonary symptoms (including dyspnoea, cough, and chest pain) and any long COVID symptoms were higher among people of North African, Middle Eastern, Eastern European, and Asian origins than among native Danes in both unadjusted and adjusted models. Despite including the nationwide sample of individuals diagnosed with COVID-19, the precision of our estimates on long COVID was limited to the sample of patients with symptoms who had contacted the hospital.

## Conclusions

Belonging to an ethnic minority group was significantly associated with an increased risk of long COVID, indicating the need to better understand long COVID drivers and address care and treatment strategies in these populations.

## Author summary

### Why was this study done?

- Evidence indicates overrepresentation of ethnic minorities among those tested positive for Severe Acute Respiratory Syndrome Coronavirus 2 (SARS-CoV-2), hospitalised for Coronavirus Disease 2019 (COVID-19), and died from COVID-19.

- After acute COVID-19 infection, many COVID-19 survivors experience a range of symptoms persisting beyond weeks or months, the condition known as long COVID.

- However, little is known about the risk of long COVID among ethnic minorities, and no existing studies had compared symptoms distribution before and after COVID-19 diagnosis in these populations.

### What did the researchers do and find?

- A nationwide register-based cohort study was performed on individuals diagnosed with COVID-19 between January 2020 and August 2022 in Denmark.

- We found that people of North African, Middle Eastern, Eastern European, and Asian origins had a higher risk of long COVID diagnosis than native Danes, with the greatest ethnic disparities being observed in the early phase of COVID-19 pandemic (January 2020 to June 2021).

- People of North African, Middle Eastern, Eastern European, and Asian origins were more likely to report cardiopulmonary symptoms (including dyspnoea, cough, and

chest pain) and *any* long COVID symptoms than native Danes, especially beyond 4 weeks to 6 months after COVID-19 diagnosis.

### What do these findings mean?

- These findings indicate the need to understand the drivers of long COVID in ethnic minorities and tailor preventive policies to their contexts.

- Efforts addressing disparities in socioeconomic conditions, advocacy activities for COVID-19 vaccines, and continuation of preventive measures may help reduce the burden of long COVID in ethnic minorities.

- The diagnosis of long COVID was limited to the sample of patients with symptoms who had contacted the hospital after acute COVID-19 infection.

## Introduction

Globally, millions of people have now been infected with Severe Acute Respiratory Syndrome Coronavirus 2 (SARS-COV-2), the virus causing Coronavirus Disease 2019 (COVID-19) [1]. Despite increased risk of hospitalisation and death in the first weeks of SARS-COV-2 infection, many COVID-19 survivors experience a range of symptoms including fatigue, cardiopulmonary symptoms (dyspnoea, cough, and chest pain), and neurological symptoms (headache, depression, and memory loss) persisting beyond weeks or months after the acute phase of COVID-19 infection, the condition known as long COVID as per National Institute for Health and Care Excellence (NICE) guidelines [2–5]. Long COVID or post-acute sequelae of COVID-19 is an emerging epidemic that is anticipated to affect the quality of life of many COVID-19 survivors [6,7]. Hence, understanding the demographic profile of long COVID sufferers is of use for planning healthcare services.

Ethnic minorities living in high-income countries have been disproportionately affected by COVID-19 in terms of infection rates, hospitalisations, and deaths [8,9]. However, studies on long COVID among ethnic minorities are few and their findings suggest that these populations exhibit a greater risk of long COVID [10–15]. For example, compared with the majority white populations in the United States and the United Kingdom, individuals who belong to black and Asian ethnicity were observed to have a higher chance of reporting long COVID symptoms after acute COVID-19 infection [10–14]. In the Netherlands, the risk of long COVID was found to be higher in patients of Surinamese, Moroccan, and Turkish origins than in those of Dutch origin [15]. Overall, the previous studies have several shortcomings, including studies were based on a single hospital setting or localised area [10,14,15], the studies did not compare symptoms distribution before and after COVID-19 diagnosis [10–15], and most of the studies were survey based [3,11,14]. In addition, comorbidities and socioeconomic factors such as income and education were not considered in some studies [10,11,14,15]. As previously described in Andersen's and Levesque's conceptual frameworks of healthcare utilisation, disparities in access to diagnosis, treatment, and care may be attributed to several factors, including individual-related factors (such as socioeconomic status, cultural beliefs, health insurance coverage, and language proficiency) and structural-related factors (such as healthcare providers' attitudes, health policy, geographical location of healthcare facility, and

availability of professional medical interpreters) [16,17]. In fact, low socioeconomic status (i.e., low income, low educational attainment, poor housing and working conditions) has previously been demonstrated to influence COVID-19 incidence and hospitalisations in ethnic minorities [18–20]. Similarly, recent evidence suggests that low socioeconomic status is significantly associated with increased risk of long COVID [21]. In Denmark, access to care including testing for COVID-19 infection is free of charge for all registered residents regardless of social position, sex, race, or ethnicity [22]. However, disparities in access to care still exist when comparing ethnic minorities and native Danes [23]. Hence, it has been reported that factors like lack of knowledge about the Danish healthcare system, language barriers, strong cultural norms, and healthcare providers' stereotypical views and cultural insensitivity affect healthcare utilisation among ethnic minorities [23]. On the other hand, evidence has emerged showing that older age, disease severity, intensive care use, comorbidities, and not receiving COVID-19 vaccine are associated with increased risk of long COVID in the general population [2,3,24]. However, it remains largely unknown to what extent these factors influence ethnic minorities' risk of long COVID.

The present study sought to address these limitations by using nationwide register data from individuals diagnosed with COVID-19 in Denmark. First, we hypothesised that ethnic minorities (defined by their region and country of origin) have a higher risk of long COVID diagnosis compared to native Danes taking into account comorbidities, socioeconomic factors, civil status, COVID-19-related hospitalisation, and vaccination status. Second, we examined whether the risk of fatigue, headache, cardiopulmonary symptoms (dyspnoea, cough, and chest pain), or *any* of these long COVID symptoms differed between ethnic minorities and native Danes within 6 months before COVID-19 diagnosis, 0 to 4 weeks, and >4 weeks to 6 months after COVID-19 diagnosis.

## Methods

### Ethics statement

This study was approved by the Danish Data Protection Agency, reference number 514-0670/21-3000. No further approval is required regarding registry-based research in Denmark.

### Setting

Denmark has a population of approximately 5.8 million people. Testing for SARS-COV-2 infection by polymerase chain reaction (PCR) was launched in March 2020. During March to May 2020, testing for SARS-COV-2 by PCR was offered for individuals with mild to severe symptoms of respiratory tract infection [25]. Universal testing for SARS-COV-2 infection by PCR was nationally implemented from May 18, 2020. Additionally, vaccination against COVID-19 started on December 27, 2020 [25]. Testing and vaccination against COVID-19 have been free of charge throughout the COVID-19 pandemic and are still free of charge for all registered residents as these services are financed by general taxes in Denmark [22].

### Data sources and study population

This nationwide register-based cohort study utilised data from the Danish COVID-19 surveillance database, the Danish National Patient Registry (DNPR), the Danish Vaccination Register, and Statistics Denmark. The study has a research protocol that was used for seeking ethical approval and gathering data from the relevant data custodians (S1 Study Protocol). The study population included all individuals residing in Denmark who had first time tested positive for SARS-CoV-2 (COVID-19 diagnosis) aged 18 years or older from January 1, 2020 to August

31, 2022 [26]. The study population was linked with the DNPR, which is a nationwide hospital register containing information on all primary and secondary diagnoses among hospitalised patients [27]. The DNPR contributed data on individuals who had COVID-19 as the primary reason for hospitalisation identified in accordance with 10th version of International Standard Classification of Diseases (ICD-10): ICD-10 codes B34.2, B34.2A, B97.2, or B97.2A. Furthermore, the DNPR provided information on comorbidities and hospital contacts related to symptoms (i.e., fatigue, headache, dyspnoea, cough, chest pain, depression, and anxiety) before and after COVID-19 diagnosis. Symptoms were recorded during hospital admissions, outpatient attendance, and attendance at emergency department. We retrieved data on first, second, and third dose of COVID-19 vaccine from the Danish Vaccination Register [28]. Statistics Denmark contributed individual-level data on country of origin, date of immigration, highest attained education, family income, civil status, and date of death [29–31]. Linkage between the registers was possible due to the availability of unique personal identification number assigned to all Danish residents [31].

## Region and country of origin

The study population was categorised based on individual and parental region and country of origin [32]. The following 8 groups were constructed according to their region of origin, with these groups being the modified version of those used by the World Bank: (i) Denmark; (ii) Northern Europe other than Denmark; (iii) Western Europe; (iv) Eastern Europe; (v) Asia; (vi) Middle East; (vii) North Africa; and (viii) sub-Saharan Africa [33]. Participants from North America, South America, and Oceania were excluded in the study as their numbers were relatively small. Furthermore, we classified the study population based on the largest countries of origin among the population of ethnic minorities residing in Denmark. The largest countries of origin selected were Norway, Sweden, Afghanistan, Iraq, Iran, Somalia, Pakistan, and Turkey. Individuals originating outside Denmark and their descendants (i.e., born in Denmark from parents with foreign citizenship) formed the ethnic minority population [32]. Participants originating and/or born in Denmark, i.e., including their descendants, constituted the reference group (native Danes). As per Statistics Denmark definition, descendants of ethnic Danes and descendants of ethnic minorities are never classified into the same group [32]. These 2 groups have different coding system based on the data from Statistics Denmark and can explicitly be separated from one another.

## Outcome

The study participants were followed up from the date of a positive test for SARS-CoV-2 infection until a long COVID diagnosis, death, emigration, or study end (August 31, 2022), whichever came first. The primary outcome of interest was ICD-10 diagnosis of long COVID identified by ICD-10 codes (B94.8 or B94.8A), and this indicates complications persisting beyond the acute COVID-19 infection that cannot be explained by an alternative diagnosis [34]. The complications encompass symptoms such as fatigue, headache, dyspnoea, chest pain, cough, and depression or anxiety that generally have an impact on everyday functioning and may present as new onset following initial recovery from an acute COVID-19 episode or persist from the initial illness [35]. The presence of a long COVID diagnosis was determined by both ICD-10 codes and the actual date of diagnosis. In addition, we examined hospital contacts related to long COVID symptoms such as fatigue, headache, dyspnoea, chest pain, cough, and depression or anxiety as a secondary outcome. Symptoms were identified by ICD-10 codes in relation to the date of hospital contact (S1 Table). Due to small outcome events on a single symptom by ethnic group, some symptoms were assessed as a composite outcome. In the

present study, the following groups of symptoms were considered: fatigue, headache, cardio-pulmonary symptoms (including dyspnoea, cough, and chest pain), and *any* of these selected long COVID symptoms (including fatigue, headache, dyspnoea, cough, chest pain, depression, and/or anxiety). We analysed the specified groups of symptoms in 3 different periods: within 6 months before COVID-19 diagnosis, 0 to 4 weeks (acute phase of COVID-19 infection), and >4 weeks to 6 months after COVID-19 diagnosis.

## Covariates

Covariates included in the analysis were age, sex, comorbidities, civil status, highest attained education, family income, length of residency, COVID-19 hospitalisation (as a proxy for disease severity), and vaccination against COVID-19. Age was analysed as a continuous variable and subsequently categorised as 18 to 60 years and >60 years in further analyses. COVID-19 hospitalisation was assessed as yes or no. Presence of comorbidities was determined by Charlson comorbidity index (CCI) based on discharge diagnosis within 5 years prior to COVID-19 diagnosis (S1 Table). The CCI included 17 diseases with scores assigned according to their severity [36]. The CCI score was divided into 3 groups: 0 (indicating no comorbidity), 1 to 2, and ≥3. Vaccination status was defined as receiving 2 doses of the COVID-19 vaccine in the analysis. Civil status was classified as cohabiting, living alone, or other. Education was grouped as low, medium, or high in accordance with the International Standard Classification of Education [37]. Family income was categorised as low, middle, or high tertiles according to the total household disposable income among patients with COVID-19 in the specific calendar year. Length of residency was a time difference in years between date of arrival in Denmark and date of COVID-19 diagnosis.

## Statistical analyses

**Analysis plan and amendments.** With an increasing number of patients attending long COVID clinics during the pandemic [6], the hypothesis of this study was that ethnic minorities have an increased risk of long COVID diagnosis than native Danes. We developed analytical plan aiming at testing this hypothesis using nationwide register data in Denmark as there were few studies using registers/electronic medical records. We had initially planned to analyse nonhospitalised individuals only, but later, we decided to include data on both hospitalised and nonhospitalised individuals after a thorough discussion among the investigators. Regarding long COVID symptoms, our original idea was to examine chances of reporting symptoms such as fatigue, headache, and cardiopulmonary symptoms within 4 weeks to 6 months after COVID-19 diagnosis among ethnic minorities compared with native Danes. However, upon reviewing the literature, we realised it would be more beneficial to investigate chances of reporting these symptoms even before COVID-19 diagnosis, and that is why we presented our analysis of symptoms within 6 months before COVID-19 diagnosis, 0 to 4 weeks, and >4 weeks to 6 months after COVID-19 diagnosis. After reviewers' and editors' input, we added analysis of risk of long COVID diagnosis by number of doses of COVID-19 vaccine and by 3 different periods of COVID-19 infection reflecting the prevalence of specific variants in Denmark by region of origin: January 2020 to June 2021 (alpha, beta, and gamma variants), July 2021 to January 2022 (delta variant), and February 2022 to August 2022 (omicron variant).

**Main analysis.** Categorical and continuous variables were summarised by frequencies and percentages and by medians and interquartile ranges, respectively. We computed age-standardised incidence rates of long COVID diagnosis per 100,000 person-years among ethnic minorities and native Danes using the 2020 Danish population as reference standard by direct method of standardisation (S1 Appendix). We used multivariable Cox proportional hazard

regression models to investigate the association between region and country of origin and the risk of long COVID diagnosis. Age, sex, civil status, education, family income, and CCI were identified as confounders using directed acyclic graphs; hence, these covariates were adjusted in the Cox models (S1 Fig). We refrained from adjusting for length of residency, COVID-19 hospitalisation, and COVID-19 vaccination status as these covariates were deemed to belong in the causal pathway for the risk of long COVID/reporting long COVID symptoms. Hence, adjusting for mediators would have induced bias as suggested in the Tutorial on directed acyclic graphs [38]. The proportional hazard assumption was assessed by Schoenfeld residuals. In addition, we performed subgroup analyses in which the hazard of long COVID diagnosis was compared between ethnic minorities and native Danes by age groups (18 to 60 years and >60 years), by COVID-19 hospitalisation (no versus yes), by COVID-19 vaccination (yes versus no), and by 3 different periods of COVID-19 infection: January 2020 to June 2021, July 2021 to January 2022, and February 2022 to August 2022. Furthermore, we assessed the association between region and country of origin and hospital contacts related to groups of symptoms by fitting multivariable logistic regression models adjusting for the same set of covariates as in the Cox models. We compared hospital contacts related to groups of symptoms within 6 months after versus 6 months before COVID-19 diagnosis in each ethnic group. Subsequently, we analysed hospital contacts related to groups of symptoms comparing ethnic minorities and native Danes in 3 time periods: 6 months before COVID-19 diagnosis, 0 to 4 weeks, and >4 weeks to 6 months after COVID-19 diagnosis. All hazard ratios (HRs) and odds ratios (ORs) with their corresponding 95% confidence interval (CI) were presented as unadjusted and adjusted, with native Danes regarded as the reference population. All analyses were performed in R statistical software (version 4.2.2). We used two-tailed tests and a $p$-value of less than 0.05 was considered statistically significant. This study was reported as per the Reporting of studies Conducted using Observational Routinely-collected health Data (RECORD) statement (S1 Checklist).

## Results

### Participants characteristics

Between January 2020 and August 2022, 2,287,175 individuals were first time diagnosed with COVID-19, of whom 39,876 (1.7%) were hospitalised and 2,247,299 (98.3%) were nonhospitalised individuals. Of the diagnosed COVID-19 cases, 1,952,021 (85.3%) were native Danes and 335,154 (14.7%) were ethnic minorities (Fig 1). Overall, 6,479 (0.3%) native Danes and 755 (0.2%) people from ethnic minorities died within 6 months after COVID-19 diagnosis.

The results on sociodemographic characteristics of the study participants showed that compared with native Danes, ethnic minorities, particularly those from Eastern Europe, Asia, Middle East, North Africa, and sub-Saharan Africa were younger at the time of COVID-19 diagnosis and were more likely to have low level of education and more likely to have low family income (Table 1). People of North African (4.6%), Middle Eastern (4.2%), Eastern European (2.7%), Asian (2.8%), and sub-Saharan African (2.5%) origin were in general more likely than native Danes (1.5%) to be hospitalised for COVID-19. Additionally, ethnic minorities from North Africa (28.5%) and Middle East (26.1%) as well as those from Pakistan (28.5%), Turkey (27.8%), Iraq (27.5%), Iran (26.2%), and Afghanistan (24.7%) had a higher proportion of individuals with comorbidities (CCI score of 1 to 2) than native Danes (20.3%) (S2 Table). We found that most ethnic minorities, especially those of North African, Middle Eastern, sub-Saharan African, and Eastern European origins had lower uptake of COVID-19 vaccine than native Danes (Table 1 and S2 Table).

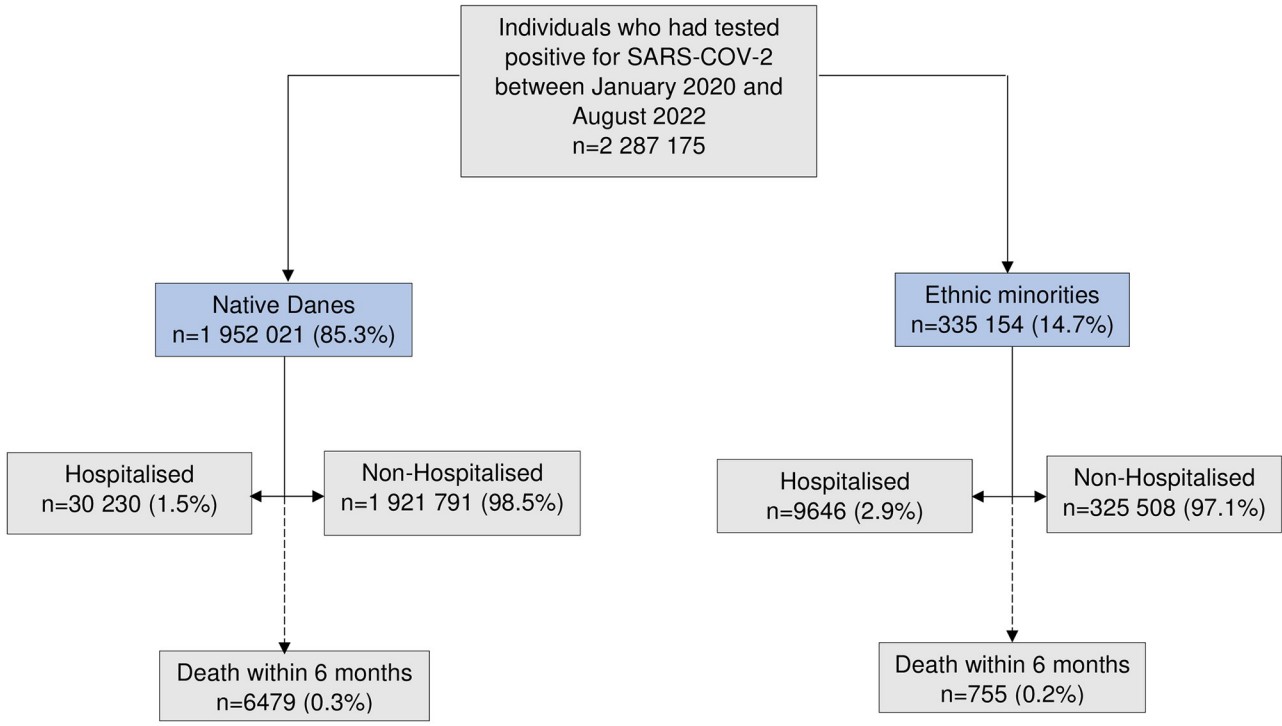

**Fig 1. Flowchart of the study population.** Ethnic minorities composed individuals originating outside Denmark (immigrants and their descendants). Native Danes composed individuals originating and/or born in Denmark, including their descendants. Death was defined as all-cause mortality within 6 months after COVID-19 diagnosis.

## Risk of long COVID diagnosis

Compared with native Danes, the greatest age-standardised incidence rate of long COVID diagnosis was observed in people of North African origin, followed by people of Middle Eastern, sub-Saharan African, Asian, and Eastern European origins (S3 Table). We found that ethnic minorities from North Africa ($n = 62$, HR 1.35, 95% CI [1.10,1.67], $p < 0.001$), Middle East ($n = 312$, HR 1.31, 95% CI [1.18,1.44], $p < 0.001$), Eastern Europe ($n = 373$, HR 1.21, 95% CI [1.11,1.32], $p < 0.001$), and Asia ($n = 204$, HR 1.14, 95% CI [1.03,1.28], $p < 0.001$) had a higher risk of long COVID diagnosis than native Danes in unadjusted model (Fig 2). After adjustment for age, sex, civil status, education, family income, and comorbidities, the risk of long COVID diagnosis remained significantly higher in people of North African (adjusted hazard ratio [aHR] 1.41, 95% CI [1.12,1.79], $p = 0.003$), Middle Eastern (aHR 1.38, 95% CI [1.24,1.55], $p < 0.001$), Eastern European (aHR 1.35, 95% CI [1.22,1.49], $p < 0.001$), and Asian (aHR 1.23, 95% CI [1.09,1.40], $p = 0.001$) origin than in native Danes. In the analysis by largest countries of origin, the results were most evident in people originating from Iraq ($n = 114$, aHR 1.56, 95% CI [1.30,1.88], $p < 0.001$), Turkey ($n = 187$, aHR 1.42, 95% CI [1.24,1.63], $p < 0.001$), and Somalia ($n = 41$, aHR 1.42, 95% CI [1.07,1.91], $p = 0.016$) (Fig 3).

When investigating factors associated with increased risk of long COVID diagnosis, we observed that compared with native Danes aged >60 years, the risk of long COVID diagnosis was highest among people of sub-Saharan African origin aged >60 years ($n = 15$, aHR 3.27, 95% CI [2.24,4.79], $p < 0.001$) (S4 and S5 Tables). Analysis by sex showed that the risk of long COVID diagnosis was significantly higher in men than in women among people of sub-Saharan African, Middle Eastern, and Eastern European origins. However, in the native Danish

**Table 1. Individuals who had first time tested positive for SARS-CoV-2 between January 2020 and August 2022 by region of origin.**

| | Denmark | Northern Europe | Western Europe | Eastern Europe | Asia | Middle East | North Africa | sub-Saharan Africa |
|---|---|---|---|---|---|---|---|---|
| **n** | 1,952,021 | 19,842 | 37,300 | 125,517 | 62,192 | 59,358 | 8,693 | 22,252 |
| **Immigrants** | NA | 18,131 (91.4%) | 34,910 (93.6%) | 100,063 (79.7%) | 49,690 (79.9%) | 45,539 (76.7%) | 5,146 (59.2%) | 18,274 (82.1%) |
| **Descendants** | NA | 1,711 (8.6%) | 2,390 (6.4%) | 25,454 (20.3%) | 12,502 (20.1%) | 13,819 (23.3%) | 3,547 (40.8%) | 3,978 (17.9%) |
| **Length of residency, years** | NA | 34 (14–39) | 28 (10–39) | 26 (13–35) | 24 (15–35) | 22 (9–30) | 31 (23–37) | 21 (12–26) |
| **Age, years** | 61 (43–75) | 60 (38–75) | 57 (38–75) | 45 (33–59) | 46 (34–60) | 45 (32–57) | 52 (36–65) | 41 (31–55) |
| **Sex** | | | | | | | | |
| Female | 1,026,373 (52.6%) | 12,471 (62.8%) | 17,288 (46.3%) | 66,216 (52.7%) | 36,541 (58.7%) | 29,022 (48.9%) | 4,386 (50.4%) | 11,524 (51.8%) |
| Male | 925,648 (47.4%) | 7,371 (37.2%) | 20,012 (53.7%) | 59,301 (47.3%) | 25,651 (41.3%) | 30,336 (51.1%) | 4,307 (49.6%) | 10,728 (48.2%) |
| **Civil status** | | | | | | | | |
| Cohabiting | 862,240 (44.2%) | 7,478 (37.7%) | 14,968 (40.1%) | 61,418 (48.9%) | 36,289 (58.3%) | 25,103 (42.3%) | 4,222 (48.6%) | 7,300 (32.8%) |
| Living alone | 852,792 (43.7%) | 10,369 (52.3%) | 19,203 (51.5%) | 53,768 (42.9%) | 20,893 (33.6%) | 28,750 (48.4%) | 3,251 (37.4%) | 12,230 (55.0%) |
| Other | 236,989 (12.1%) | 1,995 (10.0%) | 3,129 (8.4%) | 10,331 (8.2%) | 5,010 (8.1%) | 5,505 (9.3%) | 1,220 (14.0%) | 2,722 (12.2%) |
| **Education** | | | | | | | | |
| Low | 465,063 (23.8%) | 2,056 (10.4%) | 3,400 (9.1%) | 29,781 (23.7%) | 17,855 (28.7%) | 25,713 (43.3%) | 3,063 (35.2%) | 9,419 (42.3%) |
| Medium | 904,288 (46.3%) | 6,757 (34.0%) | 11,004 (29.5%) | 51,271 (40.8%) | 21,218 (34.1%) | 17,769 (29.9%) | 3,191 (36.7%) | 7,500 (33.7%) |
| High | 571,393 (29.3%) | 9,804 (49.4%) | 20,482 (54.9%) | 37,218 (29.7%) | 19,081 (30.7%) | 10,972 (18.5%) | 1,907 (22.0%) | 3,282 (15.2%) |
| Missing | 11,277 (0.6%) | 1,225 (6.2%) | 2,414 (6.5%) | 7,247 (5.8%) | 4,038 (6.5%) | 4,904 (8.3%) | 532 (6.1%) | 1,951 (8.8%) |
| **Family income*** | | | | | | | | |
| Low | 353,452 (18.1%) | 6,433 (32.4%) | 13,275 (35.6%) | 54,973 (43.8%) | 28,051 (45.1%) | 39,598 (66.7%) | 4,830 (55.6%) | 14,264 (64.1%) |
| Middle | 585,134 (30.0%) | 4,637 (23.4%) | 8,378 (22.5%) | 39,808 (31.7%) | 18,003 (28.9%) | 9,153 (15.4%) | 2,253 (25.9%) | 4,137 (18.6%) |
| High | 859,757 (44.0%) | 7,173 (36.2%) | 12,851 (34.4%) | 19,974 (15.9%) | 11,789 (19.0%) | 5,028 (8.5%) | 961 (11.0%) | 1,835 (8.2%) |
| Missing | 153,678 (7.9%) | 1,599 (8.0%) | 2,796 (7.5%) | 10,762 (8.6%) | 4,349 (7.0%) | 5,579 (9.4%) | 649 (7.5%) | 2,016 (9.1%) |
| **COVID-19 hospitalisation** | 30,230 (1.5%) | 343 (1.7%) | 543 (1.4%) | 3,423 (2.7%) | 1,798 (2.8%) | 2,551 (4.2%) | 413 (4.6%) | 575 (2.5%) |
| **Intensive care** | 12,014 (0.6%) | 100 (0.5%) | 167 (0.4%) | 504 (0.4%) | 226 (0.4%) | 280 (0.5%) | 57 (0.6%) | 148 (0.7%) |
| **COVID-19 vaccination** | | | | | | | | |
| One dose | 1,813,312 (92.9%) | 17,465 (88.0%) | 31,702 (85.0%) | 80,249 (63.9%) | 55,003 (88.4%) | 40,781 (68.7%) | 5,176 (59.5%) | 15,400 (69.2%) |
| Two doses | 1,796,381 (92.0%) | 17,104 (86.2%) | 31,027 (83.1%) | 76,751 (61.1%) | 53,820 (86.5%) | 38,722 (65.2%) | 4,924 (56.6%) | 14,483 (65.1%) |
| Three doses | 1,489,444 (76.3%) | 13,044 (65.7%) | 23,220 (62.3%) | 38,161 (30.4%) | 35,449 (57.0%) | 17,321 (29.2%) | 2,560 (29.4%) | 5,897 (26.5%) |
| **CCI§** | | | | | | | | |
| 0 | 1,550,412 (79.4%) | 16,412 (82.7%) | 31,994 (85.8%) | 101,384 (80.8%) | 50,558 (81.3%) | 43,825 (73.8%) | 6,205 (71.4%) | 18,041 (81.1%) |
| 1–2 | 396,200 (20.3%) | 3,381 (17.0%) | 5,234 (14.0%) | 23,972 (19.1%) | 11,563 (18.6%) | 15,470 (26.1%) | 2,475 (28.5%) | 4,154 (18.7%) |
| ≥3 | 5,409 (0.3%) | 49 (0.3%) | 72 (0.2%) | 161 (0.1%) | 71 (0.1%) | 63 (0.1%) | 13 (0.1%) | 57 (0.2%) |

Data are in median (IQR) or n (%). Northern Europe indicates Northern Europe other than Denmark.

*Family income (presented in tertiles) was the total household disposable income among patients with COVID-19 in the specific calendar year.

§CCI composed myocardial infarction, congestive heart failure, peripheral vascular disease, cerebrovascular disease, chronic obstructive pulmonary disease, rheumatic disease, dementia, peptic ulcer disease, hemiplegia, diabetes without complications, diabetes with complications, mild liver disease, moderate to severe liver disease, renal disease, malignancy, metastatic cancer, and AIDS.

AIDS, acquired immunodeficiency syndrome; CCI, Charlson comorbidity index; COVID-19, Coronavirus Disease 2019; IQR, interquartile range; NA, not applicable; SARS-CoV-2, Severe Acute Respiratory Syndrome Coronavirus 2.

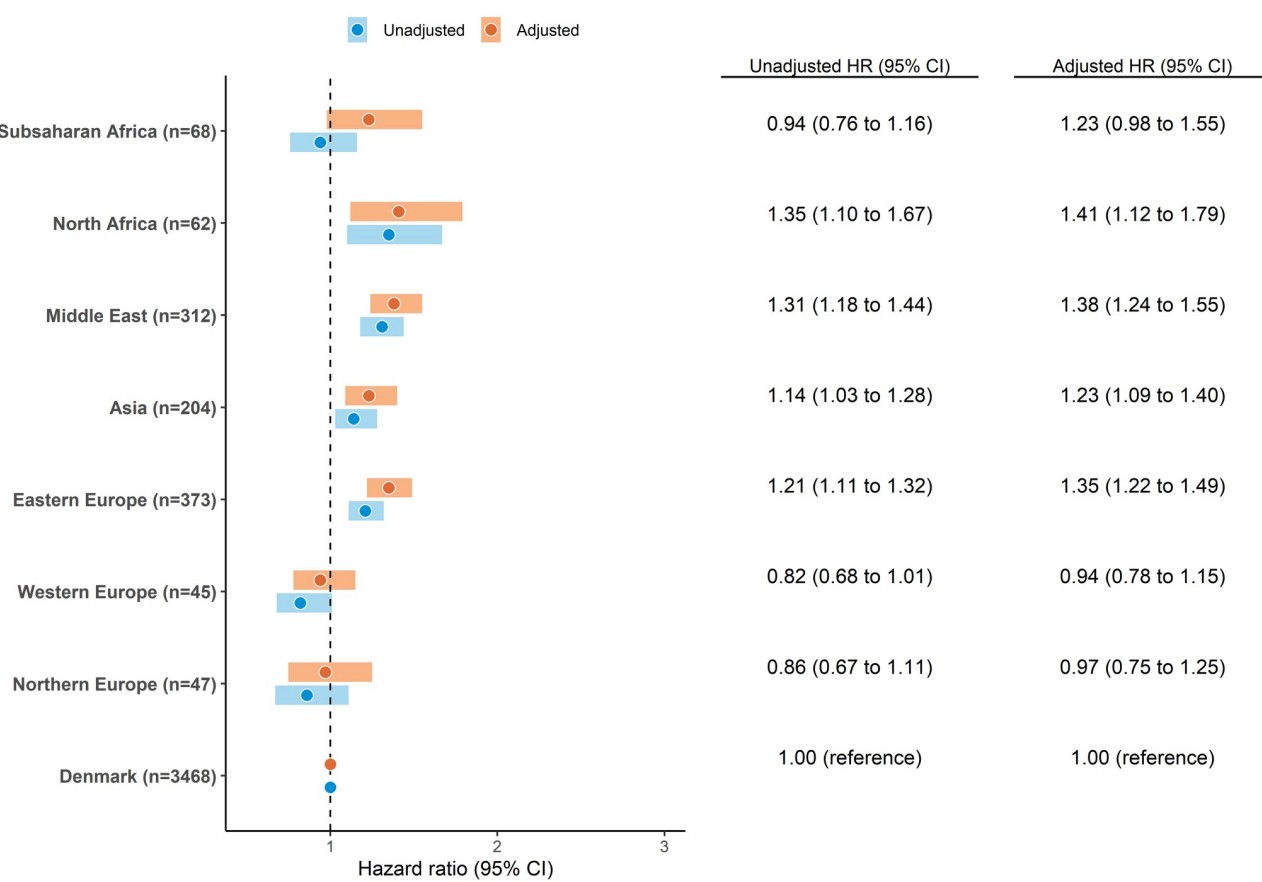

**Fig 2. Hazard ratios of long COVID diagnosis by region of origin.** Northern Europe indicates Northern Europe other than Denmark. The adjusted model composed age, sex, civil status, education, family income, and CCI. CCI, Charlson comorbidity index; CI, confidence interval; HR, hazard ratio.

population, the risk of long COVID diagnosis was significantly lower among men than among women (S6 Table). Moreover, in the men population, we found that compared with native Danish men, the risk of long COVID diagnosis was higher in men from North Africa, Middle East, sub-Saharan Africa, Asia, and Eastern Europe. While no ethnic differences in long COVID risk among the population of women were observed (S7 Table).

Compared with nonhospitalised native Danes, COVID-19 hospitalisation was significantly associated with a higher risk of long COVID diagnosis among both native Danes and ethnic minorities before and after adjustment for confounders (Table 2). However, the HRs of long COVID diagnosis for ethnic minorities from North Africa ($n = 33$, aHR 3.98, 95% CI [2.75,5.75], $p < 0.001$), Middle East ($n = 161$, aHR 4.43, 95% CI [3.71,5.29], $p < 0.001$), Eastern Europe ($n = 204$, aHR 4.49, 95% CI [3.84,5.23], $p < 0.001$), Asia ($n = 108$, aHR 3.44, 95% CI [2.79,4.23], $p < 0.001$), and sub-Saharan Africa ($n = 37$, aHR 4.30, 95% CI [3.05,6.07], $p < 0.001$) were still higher than that of native Danes ($n = 1,483$, aHR 2.82, 95% CI [2.64,3.00], $p<0.001$) among individuals hospitalised for COVID-19. Among the nonhospitalised individuals, people of Eastern European and Middle Eastern origins were the groups that had a higher risk of long COVID diagnosis than native Danes (S8 Table). Further analysis showed that individuals not receiving COVID-19 vaccine exhibited greater risk of long COVID diagnosis than

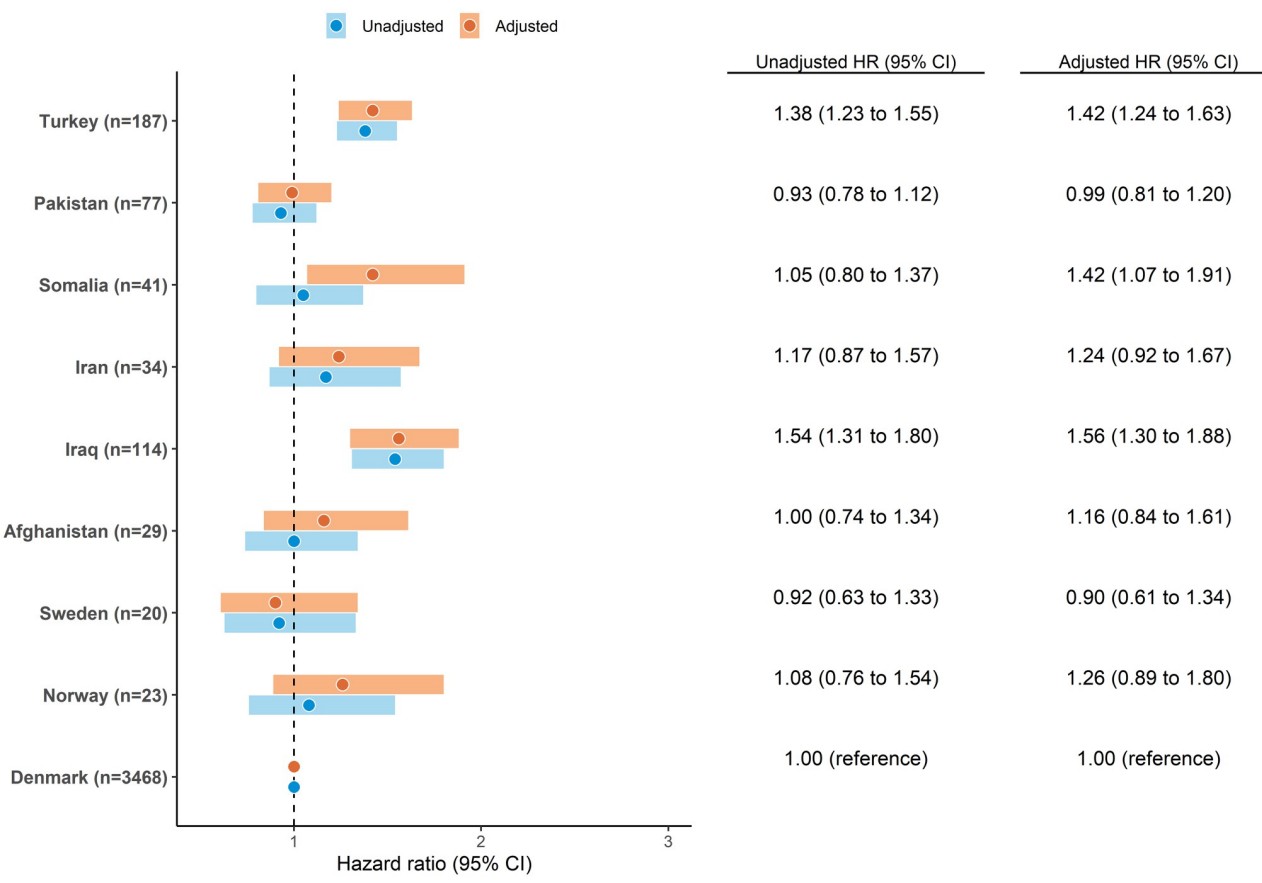

**Fig 3. Hazard ratios of long COVID diagnosis by largest countries of origin.** The adjusted model composed age, sex, civil status, education, family income, and CCI. CCI, Charlson comorbidity index; CI, confidence interval; HR, hazard ratio.

individuals vaccinated, and the association was found in native Danes only (aHR 1.47, 95% CI [1.33,1.63], *p* for interaction <0.001). Similarly, analysis by number of doses of vaccination revealed that native Danes only had significantly reduced risk of long COVID diagnosis with receiving 2 and 3 doses of COVID-19 vaccine as compared to not receiving vaccination (S9 Table). Overall, large ethnic disparities in the risk of long COVID diagnosis were observed between January 2020 and June 2021 as compared to other periods of COVID-19 infection during pandemic (Table 3).

## Hospital contacts related to long COVID symptoms

The majority of ethnic minority groups and native Danes exhibited higher odds of hospital contacts related to fatigue, headache, cardiopulmonary symptoms, and *any* long COVID symptoms within 6 months after COVID-19 diagnosis as compared to 6 months before COVID-19 diagnosis in the adjusted estimates (Fig 4 and S10 Table). However, compared with native Danes, differences in ORs of hospital contacts related to cardiopulmonary symptoms and *any* long COVID symptoms were more pronounced among people of North African, Middle Eastern, Eastern European, Asian, and Northern European origins, especially beyond 4 weeks to 6 months after COVID-19 diagnosis in both unadjusted and adjusted estimates

**Table 2. Hazard ratios of long COVID diagnosis by hospitalisation for COVID-19.**

| | COVID-19 hospitalisation | n | Unadjusted HR (95% CI) | Adjusted HR (95% CI) |
|---|---|---|---|---|
| Denmark | No | 1,985 | 1.00 (reference) | 1.00 (reference) |
| | Yes | 1,483 | 6.53 (6.14 to 6.95) | 2.82 (2.64 to 3.00) |
| Northern Europe | No | 25 | 0.79 (0.58 to 1.07) | 0.90 (0.66 to 1.23) |
| | Yes | 22 | 7.58 (4.98 to 11.52) | 3.44 (2.21 to 5.34) |
| Western Europe | No | 22 | 0.59 (0.43 to 0.81) | 0.73 (0.53 to 1.01) |
| | Yes | 23 | 6.15 (4.08 to 9.28) | 2.57 (1.69 to 3.92) |
| Eastern Europe | No | 169 | 1.05 (0.94 to 1.17) | 1.15 (1.02 to 1.30) |
| | Yes | 204 | 10.67 (9.27 to 12.29) | 4.49 (3.84 to 5.23) |
| Asia | No | 96 | 1.04 (0.90 to 1.20) | 1.14 (0.98 to 1.33) |
| | Yes | 108 | 8.83 (7.29 to 10.70) | 3.44 (2.79 to 4.23) |
| Middle East | No | 151 | 1.14 (1.01 to 1.29) | 1.21 (1.05 to 1.39) |
| | Yes | 161 | 10.16 (8.67 to 11.90) | 4.43 (3.71 to 5.29) |
| North Africa | No | 29 | 1.25 (0.96 to 1.63) | 1.27 (0.94 to 1.71) |
| | Yes | 33 | 8.31 (5.89 to 11.73) | 3.98 (2.75 to 5.75) |
| sub-Saharan Africa | No | 31 | 0.77 (0.58 to 1.01) | 0.99 (0.73 to 1.33) |
| | Yes | 37 | 8.48 (6.13 to 11.73) | 4.30 (3.05 to 6.07) |

Northern Europe indicates Northern Europe other than Denmark. The adjusted model composed age, sex, civil status, education, family income, and CCI.

CCI, Charlson comorbidity index; CI, confidence interval; COVID-19, Coronavirus Disease 2019; HR, hazard ratio.

(Table 4). Although people of sub-Saharan African origin did not show significant difference from native Danes in the odds of hospital contacts related to *any* long COVID symptoms, this group was observed to have higher odds of hospital contacts related to symptoms like fatigue, headache, and cardiopulmonary symptoms beyond 4 weeks to 6 months after COVID-19 diagnosis. Moreover, analysis by largest countries of origin revealed that ethnic minority groups, especially those of Swedish, Afghan, Iraqi, Iranian, Somali, Pakistani, and Turkish origins had higher odds of hospital contacts related to *any* long COVID symptoms than native Danes, particularly beyond 4 weeks to 6 months after COVID-19 diagnosis in both unadjusted and adjusted models (S11 and S12 Tables).

**Table 3. Hazard ratios of long COVID diagnosis in 3 periods of COVID-19 infection by region of origin.**

| | January 2020 to June 2021 (alpha, beta, and gamma variants period) | | July 2021 to January 2022 (delta variant period) | | February 2022 to August 2022 (omicron variant period) | |
|---|---|---|---|---|---|---|
| | n | Adjusted HR (95% CI) | n | Adjusted HR (95% CI) | n | Adjusted HR (95% CI) |
| Denmark | 1,761 | 1.00 (reference) | 883 | 1.00 (reference) | 824 | 1.00 (reference) |
| Northern Europe | 22 | 0.95 (0.64 to 1.41) | 14 | 0.92 (0.59 to 1.42) | 11 | 1.04 (0.61 to 1.76) |
| Western Europe | 23 | 0.74 (0.50 to 1.09) | 11 | 0.76 (0.48 to 1.22) | 11 | 1.05 (0.64 to 1.73) |
| Eastern Europe | 186 | 1.57 (1.35 to 1.83) | 110 | 1.29 (1.08 to 1.53) | 77 | 1.14 (0.93 to 1.40) |
| Asia | 121 | 1.47 (1.22 to 1.77) | 47 | 1.04 (0.81 to 1.35) | 36 | 1.26 (0.98 to 1.61) |
| Middle East | 153 | 1.74 (1.47 to 2.08) | 91 | 1.18 (0.96 to 1.44) | 68 | 1.17 (0.93 to 1.48) |
| North Africa | 30 | 1.44 (1.03 to 2.07) | 13 | 0.61 (0.33 to 1.11) | 19 | 2.30 (1.60 to 3.30) |
| sub-Saharan Africa | 35 | 1.85 (1.37 to 2.52) | 22 | 1.19 (0.80 to 1.76) | 11 | 0.48 (0.24 to 0.92) |

Northern Europe indicates Northern Europe other than Denmark. The adjusted model composed age, sex, civil status, education, family income, and CCI.

CCI, Charlson comorbidity index; CI, confidence interval; COVID-19, Coronavirus Disease 2019; HR, hazard ratio.

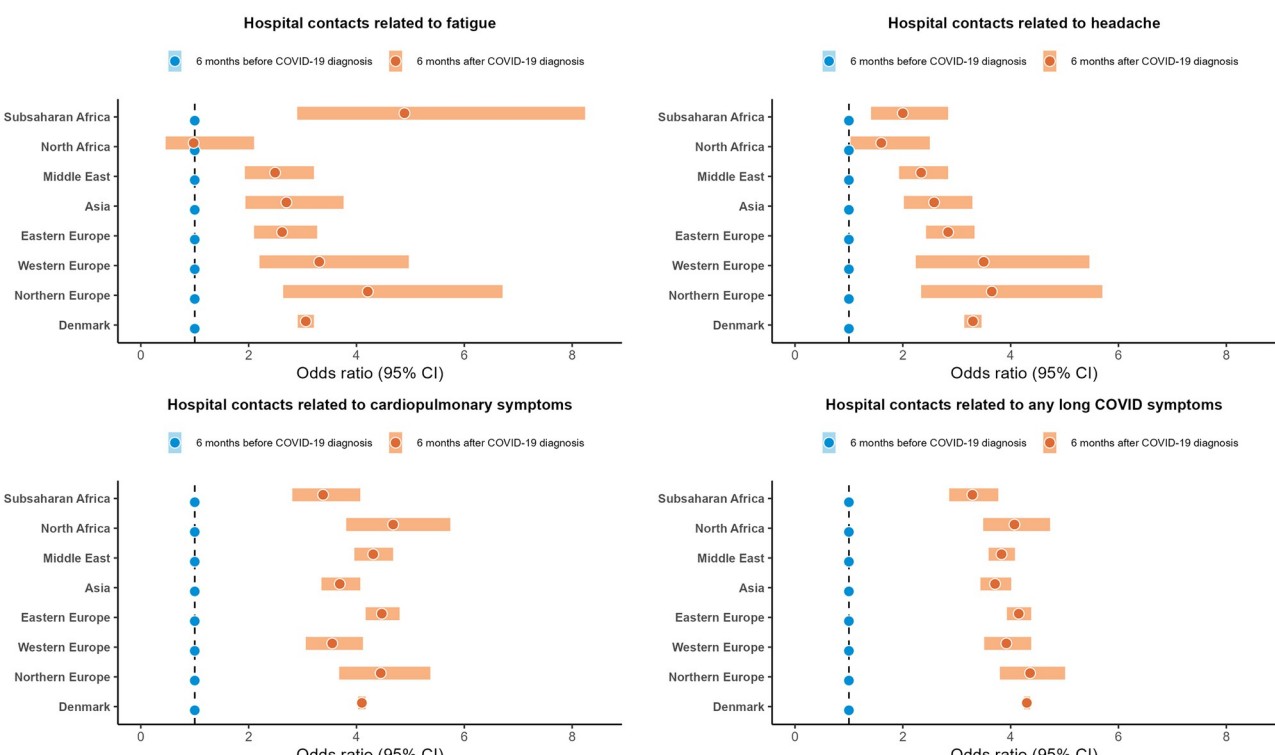

**Fig 4. Adjusted odds ratios of hospital contacts related to specific symptoms within 6 months after COVID-19 diagnosis compared with 6 months before COVID-19 diagnosis (reference group) by region of origin.** Northern Europe indicates Northern Europe other than Denmark. Hospital contacts related to cardiopulmonary symptoms included dyspnoea (difficulty in breathing), cough, and chest pain as a composite outcome. Hospital contacts related to any long COVID symptoms included fatigue, headache, dyspnoea (difficulty in breathing), cough, chest pain, depression, and/or anxiety as a composite outcome. The adjusted model composed age, sex, civil status, education, family income, and CCI. CCI, Charlson comorbidity index; CI, confidence interval.

## Discussion

This Danish nationwide cohort study found that compared with native Danes, the risk of long COVID diagnosis was higher in the majority of ethnic minority populations, notably for people of North African, Middle Eastern, Eastern European, and Asian origins in both unadjusted and adjusted models. Our findings also confirm that the chances of reporting cardiopulmonary symptoms (including dyspnoea, cough, and chest pain) and *any* long COVID symptoms were higher among people of North African, Middle Eastern, Eastern European, and Asian origins than among native Danes, especially beyond 4 weeks to 6 months after COVID-19 diagnosis in both unadjusted and adjusted models. In the analysis by largest countries of origin, this study found that the risk of long COVID diagnosis was higher in ethnic minorities from Iraq, Turkey, and Somalia than in native Danes after adjustment for all relevant covariates. While chances of reporting *any* long COVID symptoms were higher in people of Swedish, Afghan, Iraqi, Iranian, Somali, Pakistani, and Turkish origins than in native Danes, particularly beyond 4 weeks to 6 months after COVID-19 diagnosis in both unadjusted and adjusted estimates.

Compared with previous research, this study adds findings on the established diagnosis of long COVID among ethnic minorities living in Denmark. To our knowledge, no previous study on long COVID in ethnic minorities that had included symptoms experienced before COVID-19 diagnosis. Overall, studies on long COVID among ethnic minorities are still

**Table 4. Odds ratios of hospital contacts related to specific symptoms by region of origin.**

| | 6 months before COVID-19 diagnosis | | | 0 to 4 weeks after COVID-19 diagnosis | | | >4 weeks to 6 months after COVID-19 diagnosis | | |
|---|---|---|---|---|---|---|---|---|---|
| | n | Unadjusted OR (95% CI) | Adjusted OR (95% CI) | n | Unadjusted OR (95% CI) | Adjusted OR (95% CI) | n | Unadjusted OR (95% CI) | Adjusted OR (95% CI) |
| **Hospital contacts related to fatigue** | | | | | | | | | |
| Denmark | 2,004 | 1.00 (reference) | 1.00 (reference) | 561 | 1.00 (reference) | 1.00 (reference) | 969 | 1.00 (reference) | 1.00 (reference) |
| Northern Europe | 19 | 0.87 (0.63 to 1.21) | 0.94 (0.67 to 1.33) | 10 | 1.13 (0.67 to 1.92) | 1.06 (0.58 to 1.93) | 15 | 1.74 (1.23 to 2.45) | 1.87 (1.30 to 2.69) |
| Western Europe | 36 | 0.91 (0.71 to 1.18) | 0.99 (0.74 to 1.31) | 16 | 1.63 (1.14 to 2.31) | 1.82 (1.27 to 2.62) | 13 | 0.62 (0.40 to 1.00) | 0.81 (0.51 to 1.28) |
| Eastern Europe | 137 | 1.00 (0.87 to 1.15) | 1.15 (0.98 to 1.34) | 43 | 1.25 (1.01 to 1.57) | 1.37 (1.05 to 1.78) | 96 | 1.41 (1.19 to 1.68) | 1.42 (1.16 to 1.74) |
| Asia | 56 | 0.83 (0.67 to 1.02) | 0.99 (0.79 to 1.25) | 15 | 1.15 (0.83 to 1.59) | 1.40 (0.97 to 2.00) | 36 | 1.08 (0.83 to 1.42) | 1.26 (0.84 to 1.69) |
| Middle East | 119 | 1.33 (1.14 to 1.56) | 1.48 (1.24 to 1.77) | 33 | 1.48 (1.13 to 1.94) | 1.68 (1.25 to 2.27) | 61 | 1.48 (1.19 to 1.83) | 1.44 (1.13 to 1.85) |
| North Africa | 19 | 1.44 (1.01 to 2.07) | 1.53 (1.04 to 2.24) | § | § | § | 10 | 1.14 (0.63 to 2.07) | 1.04 (0.54 to 2.01) |
| sub-Saharan Africa | 28 | 1.07 (0.78 to 1.47) | 0.96 (0.64 to 1.44) | 15 | 1.75 (1.11 to 2.75) | 2.32 (1.42 to 3.77) | 19 | 1.91 (1.35 to 2.71) | 2.00 (1.35 to 2.96) |
| **Hospital contacts related to headache** | | | | | | | | | |
| Denmark | 2,765 | 1.00 (reference) | 1.00 (reference) | 512 | 1.00 (reference) | 1.00 (reference) | 1630 | 1.00 (reference) | 1.00 (reference) |
| Northern Europe | 34 | 1.24 (0.94 to 1.64) | 1.04 (0.75 to 1.44) | 8 | 1.07 (0.55 to 2.07) | 1.19 (0.61 to 2.30) | 17 | 1.56 (1.12 to 2.16) | 1.73 (1.23 to 2.42) |
| Western Europe | 38 | 0.69 (0.51 to 0.92) | 0.80 (0.58 to 1.10) | 10 | 0.97 (0.56 to 1.68) | 1.21 (0.70 to 2.10) | 22 | 0.95 (0.68 to 1.32) | 1.08 (0.75 to 1.56) |
| Eastern Europe | 276 | 1.99 (1.80 to 2.19) | 1.39 (1.24 to 1.56) | 60 | 2.39 (1.95 to 2.93) | 1.68 (1.34 to 2.11) | 183 | 2.44 (2.16 to 2.75) | 1.66 (1.44 to 1.91) |
| Asia | 117 | 1.57 (1.35 to 1.83) | 1.18 (1.00 to 1.40) | 26 | 1.70 (1.23 to 2.35) | 1.08 (0.74 to 1.58) | 78 | 2.04 (1.71 to 4.59) | 1.46 (1.19 to 1.79) |
| Middle East | 206 | 2.51 (2.24 to 2.82) | 1.60 (1.40 to 1.83) | 42 | 2.38 (1.84 to 3.09) | 1.34 (0.99 to 1.80) | 121 | 2.48 (2.12 to 2.89) | 1.52 (1.27 to 1.81) |
| North Africa | 37 | 3.47 (2.75 to 4.37) | 2.27 (1.75 to 2.95) | 8 | 2.60 (1.43 to 4.72) | 1.21 (0.57 to 2.56) | 17 | 2.23 (1.51 to 3.28) | 1.53 (1.02 to 2.32) |
| sub-Saharan Africa | 48 | 2.56 (2.08 to 3.14) | 2.04 (1.64 to 2.53) | 11 | 2.44 (1.53 to 3.90) | 1.07 (0.58 to 1.96) | 39 | 2.75 (2.11 to 3.59) | 1.67 (1.22 to 2.26) |
| **Hospital contacts related to cardiopulmonary symptoms** | | | | | | | | | |
| Denmark | 14,019 | 1.00 (reference) | 1.00 (reference) | 4117 | 1.00 (reference) | 1.00 (reference) | 9027 | 1.00 (reference) | 1.00 (reference) |
| Northern Europe | 117 | 0.82 (0.72 to 0.92) | 0.80 (0.70 to 0.92) | 34 | 0.60 (0.46 to 0.78) | 0.66 (0.50 to 0.86) | 102 | 1.16 (1.02 to 1.33) | 1.33 (1.16 to 1.53) |
| Western Europe | 217 | 0.89 (0.81 to 0.98) | 1.01 (0.92 to 1.12) | 68 | 1.30 (1.13 to 1.49) | 1.57 (1.36 to 1.81) | 131 | 0.82 (0.72 to 0.93) | 0.97 (0.85 to 1.11) |
| Eastern Europe | 983 | 1.09 (1.04 to 1.14) | 1.09 (1.03 to 1.15) | 447 | 1.70 (1.58 to 1.82) | 1.87 (1.73 to 2.02) | 836 | 1.57 (1.50 to 1.66) | 1.52 (1.43 to 1.61) |
| Asia | 486 | 1.09 (1.02 to 1.17) | 1.19 (1.11 to 1.28) | 178 | 1.35 (1.21 to 1.50) | 1.48 (1.32 to 1.67) | 387 | 1.53 (1.42 to 1.64) | 1.55 (1.44 to 1.68) |
| Middle East | 720 | 1.42 (1.34 to 1.49) | 1.29 (1.21 to 1.37) | 287 | 2.07 (1.91 to 2.25) | 2.02 (1.84 to 2.22) | 539 | 1.87 (1.76 to 1.98) | 1.66 (1.54 to 1.78) |
| North Africa | 108 | 1.17 (1.02 to 1.35) | 1.07 (0.92 to 1.26) | 46 | 1.65 (1.32 to 2.05) | 1.51 (1.18 to 1.92) | 103 | 2.15 (1.87 to 2.46) | 1.93 (1.67 to 2.25) |
| sub-Saharan Africa | 168 | 1.06 (0.95 to 1.19) | 1.11 (0.97 to 1.26) | 50 | 1.12 (0.91 to 1.37) | 1.21 (0.97 to 1.52) | 122 | 1.27 (1.11 to 1.46) | 1.26 (1.08 to 1.46) |
| **Hospital contacts related to any long COVID symptoms** | | | | | | | | | |
| Denmark | 25,375 | 1.00 (reference) | 1.00 (reference) | 6506 | 1.00 (reference) | 1.00 (reference) | 17,516 | 1.00 (reference) | 1.00 (reference) |
| Northern Europe | 233 | 0.91 (0.83 to 1.00) | 0.90 (0.81 to 1.00) | 54 | 0.75 (0.62 to 0.90) | 0.83 (0.68 to 1.01) | 204 | 1.20 (1.09 to 1.33) | 1.36 (1.23 to 1.51) |
| Western Europe | 410 | 0.92 (0.85 to 1.00) | 1.05 (0.98 to 1.14) | 120 | 1.31 (1.16 to 1.49) | 1.52 (1.32 to 1.71) | 273 | 0.90 (0.82 to 0.98) | 1.10 (1.01 to 1.20) |
| Eastern Europe | 1,925 | 1.18 (1.14 to 1.23) | 1.07 (1.03 to 1.11) | 649 | 1.63 (1.54 to 1.73) | 1.64 (1.53 to 1.75) | 1,587 | 1.55 (1.49 to 1.61) | 1.33 (1.28 to 1.39) |
| Asia | 889 | 1.11 (1.05 to 1.16) | 1.07 (1.02 to 1.13) | 265 | 1.36 (1.25 to 1.49) | 1.34 (1.21 to 1.48) | 722 | 1.53 (1.45 to 1.61) | 1.34 (1.27 to 1.43) |

*(Continued)*

**Table 4.** (Continued)

| | 6 months before COVID-19 diagnosis | | | 0 to 4 weeks after COVID-19 diagnosis | | | >4 weeks to 6 months after COVID-19 diagnosis | | |
|---|---|---|---|---|---|---|---|---|---|
| | n | Unadjusted OR (95% CI) | Adjusted OR (95% CI) | n | Unadjusted OR (95% CI) | Adjusted OR (95% CI) | n | Unadjusted OR (95% CI) | Adjusted OR (95% CI) |
| Middle East | 1,382 | 1.54 (1.48 to 1.60) | 1.20 (1.14 to 1.26) | 428 | 1.97 (1.84 to 2.11) | 1.72 (1.59 to 1.86) | 1009 | 1.80 (1.72 to 1.88) | 1.31 (1.24 to 1.39) |
| North Africa | 225 | 1.43 (1.30 to 1.59) | 1.17 (1.04 to 1.30) | 64 | 1.50 (1.24 to 1.81) | 1.26 (1.03 to 1.56) | 195 | 2.13 (1.93 to 2.36) | 1.71 (1.53 to 1.91) |
| sub-Saharan Africa | 320 | 1.14 (1.05 to 1.24) | 1.03 (0.94 to 1.14) | 95 | 1.26 (1.08 to 1.47) | 1.21 (1.02 to 1.44) | 245 | 1.32 (1.20 to 1.46) | 1.11 (0.99 to 1.24) |

Northern Europe indicates Northern Europe other than Denmark. Hospital contacts related to cardiopulmonary symptoms included dyspnoea (difficulty in breathing), cough, and chest pain as a composite outcome. Hospital contacts related to any long COVID symptoms included fatigue, headache, dyspnoea (difficulty in breathing), cough, chest pain, depression, and/or anxiety as a composite outcome.

§Estimates are not displayed due to small numbers in accordance with Danish Data Protection Act.

The adjusted model composed age, sex, civil status, education, family income, and CCI.

CCI, Charlson comorbidity index; CI, confidence interval; COVID-19, Coronavirus Disease 2019; OR, odds ratio.

lacking, and this study is among the few to contribute knowledge to the existing body of literature. In line with our findings, previous studies in the US, UK, and the Netherlands have reported that people of African, Asian, and Turkish origins exhibited higher chances of reporting long COVID symptoms than the native majority population [10–15].

The observed higher risk of long COVID among ethnic minorities living in Denmark may be explained by several factors including individual-related factors (for instance, poor working and living conditions, cultural beliefs, health-seeking behaviours, and mistrust of healthcare system), structural-related factors (for instance, lack of health information in ethnic minorities' native languages and absence of professional medical interpreters), and markers of COVID-19 severity. First, it has been reported that ethnic minorities, especially those of non-Western origin, are heavily represented in certain sectors of the Danish economy such as transportation, cleaning services, and social support services [39]. These sectors require extensive interaction with the public as a result may have predisposed them to contracting COVID-19 infection in the first phase. Hence, these working conditions can as well contribute to the probability of COVID-19 reinfection if preventive measures are not in place leading to the increased risk of long COVID in ethnic minorities [21]. Second, living in a crowded area and/or sharing a single dwelling with family members of multiple generations may also have contributed to the increased risk of long COVID in ethnic minorities [18,21]. Living in these circumstances may sometimes be culturally influenced but can have detrimental effect in individual's recovery from COVID-19 infection [40]. For instance, these living conditions may have created barriers to adoption of preventive measures (for instance, self-isolation) among infected family members leading to continuation of symptoms or new onset of cardiopulmonary and neurological symptoms post-acute COVID-19 infection. Additionally, cultural beliefs that are embedded in certain cultures may have partly influenced the risk of long COVID in ethnic minorities. The influence of culture on health is generally vast as it may affect individuals' perception of diseases, causes of diseases, and approaches to health promotion and preventive measures [40]. Another factor that may be contributing to the increased risk of long COVID diagnosis among ethnic minorities may be attributed to their pattern of healthcare utilisation (health-seeking behaviours) prior to COVID-19 era. In further analyses, we observed that people of North African and Middle Eastern origins were more likely to contact hospital in relation to fatigue, headache, and cardiopulmonary symptoms and were more likely to

contact their own general practitioner than natives Danes even before COVID-19 pandemic (S2 Fig and S13 Table). Moreover, previous studies have also illustrated that compared with native Danes, most ethnic minorities originating from non-Western countries have increased contacts to emergency room, general practitioner, and specialist [41,42]. Hence, this pattern of healthcare utilisation may be a possible explanation for their increased risk of long COVID diagnosis. On the other hand, it is important to point out that pathways to the increased risk of long COVID among ethnic minorities may, to a certain degree, vary from one country to another due to differences in health policies, health insurance coverage, and healthcare system organisation. For example, in the US, lack of health insurance is reported to be associated with poor healthcare utilisation among ethnic minorities [18], which may be seen as a contributing factor to long COVID risk due to delays in seeking care and right health information when having acute COVID-19 infection. On the contrary, the Danish healthcare system is a residence-based system; thus, if a person is living in Denmark and registered in the Danish Civil Registration System, that person is automatically entitled to free access to hospital care, emergency care, and a general practitioner. Therefore, it is unlikely that the risk of long COVID observed in ethnic minorities is contributed by insurance coverage or health policy available. However, it is possible that factors such as lack of health information in their own native languages (as most health information is in Danish language), absence of professional medical interpreters, previous negative healthcare experience, and mistrust of healthcare system may have played a role in the risk of long COVID in some ethnic minorities [23].

Despite being the mediating factors in the casual pathway, markers of COVID-19 severity such as COVID-19 hospitalisation and intensive care use may also explain the increased risk of long COVID in ethnic minorities. Our estimates demonstrate that COVID-19 hospitalisation was associated with increased risk of long COVID in both ethnic minorities and native Danes. Notwithstanding the increased risk of long COVID in native Danes hospitalised for COVID-19, the present study found that ethnic minorities were more likely than native Danes to be hospitalised for COVID-19. In addition, differences in HRs of long COVID diagnosis were more pronounced for people of North African, Middle Eastern, Eastern European, Asian, and sub-Saharan African origins than for native Danes hospitalised for COVID-19. After removing the effect of hospitalisation, the risk of long COVID diagnosis was still higher among people of Eastern European and Middle Eastern origins than among native Danes. Hence, this may signify a greater burden of long COVID in these ethnic minority groups. Furthermore, the use of intensive care may partly contribute to the increased risk of long COVID among ethnic minorities. A recent Danish study using data from patients hospitalised for COVID-19 has reported that ethnic minorities originating from non-Western countries had a higher chance than native Danes of use of mechanical ventilation [43], which could be seen as another marker of COVID-19 severity associated with high burden of long COVID among ethnic minorities living in Denmark. Despite considering a wide range of comorbidities and socioeconomic factors (i.e., family income and education) in our models, the risk of long COVID remained significantly higher in most ethnic minorities. Therefore, this work suggests that the high burden of long COVID in these populations may also be rooted in the complex interplay between the biological factors such as COVID-19 variant and immunological factors and nonbiological factors such as barriers to accessing healthcare, differences in healthcare demand, and late contact with healthcare system when having COVID-19 infection [44]. Additionally, our findings show large ethnic disparities in the risk of long COVID diagnosis during the early phase of pandemic (i.e., January 2020 to June 2021) as compared to the latter phases. Although alpha, beta, and gamma variants were the most prevalent COVID-19 variants in Denmark during this period [45], we are unable to clearly state the influence of the specific COVID-19 variants in the increased risk of long COVID in ethnic minorities as such data were unavailable. It is

also important to highlight that in the early phase of the pandemic, the healthcare system had less knowledge and experience on how to treat the COVID-19 infection, which has most likely contributed to more pronounced ethnic disparities in long COVID in this phase. Recent evidence suggests that individuals not receiving COVID-19 vaccine have a higher risk of long COVID in the general population [24]. Although most ethnic minorities were less likely than native Danes to receive COVID-19 vaccine, their risk of long COVID did not seem to be influenced by different number of doses of vaccination.

The present study has several strengths, including using a nationwide sample of individuals diagnosed with COVID-19 in Denmark, using an established ICD-10-based diagnosis of long COVID in Denmark, and incorporating a wide range of comorbidities and sociodemographic factors. However, there are some limitations. First, long COVID diagnosis in the registers was implemented from April 2020, which entails lack of registration of long COVID cases between January and March 2020 [34]. The registration of long COVID diagnosis was based on hospital contacts related to hospital admission, contacts at emergency department, and contacts at outpatient clinic. Hence, individuals with long COVID symptoms who had contacted the general practitioner only were most likely not captured in this study. Additionally, in a hospital setting, the registration of long COVID diagnosis might be problematic due to similarities of symptoms such as dyspnoea, chest pain, and cough, which may also be experienced in other established differential diagnoses. Second, symptoms included in the study were in connection with hospital contact and identified by ICD-10 codes, which may have introduced some selection bias. This is because these symptoms may not be representative of long COVID situation in Denmark as some may choose not to contact hospital if the symptoms are not interfering with their daily routines. Therefore, it is possible that individuals are experiencing long COVID symptoms more than what is reported. Another limitation was that we considered a single long COVID–related symptom such as fatigue or headache as "long COVID" as per NICE guidelines stating that any sign and symptom that continue or develop after acute COVID-19 infection (from 4 weeks or more) are considered long COVID if not explained by an alternative diagnosis [5]. Although we adjusted for 17 preexisting health conditions using CCI when estimating ORs of reporting long COVID symptoms, there is still a possibility that these symptoms might be unrelated to COVID-19/long COVID, and, hence, this may have resulted in bias on the estimates reported. Owing to a few observations in the descendant group, we could not explore intergenerational differences in the risk of long COVID. In addition, due to low count in some ethnic groups, we could not perform analysis of a single symptom by largest countries of origin. Furthermore, data on medication used at hospitals are currently not available in the Danish registers, and that is why we could not investigate the influence of medication in the risk of long COVID comparing ethnic minorities and native Danes. Due to overlapping CI for the ORs of fatigue and headache among ethnic groups during the COVID-19 time periods (i.e., 6 months before COVID-19 diagnosis versus 0 to 4 weeks versus >4 weeks to 6 months after COVID-19 diagnosis), a caution is needed when interpreting these findings as the differences may be minor or nonexistent. Finally, we acknowledge that when estimating the risk of long COVID diagnosis/symptoms, we did not take into account some other aspects of disparities, which are connected to culture, behavioural factors, healthcare providers attitude and stereotypes, or mistrust of healthcare system as such data were unavailable.

The results of this Danish study could be applicable to other neighbouring Nordic countries because of the similarities in the composition of ethnic minorities and similarities in terms of healthcare models, which are based on the principle of free access to care for all residents regardless of socioeconomic status, race, or ethnicity. Overall, the applicability of these findings beyond the Nordic countries may be somewhat difficult due to differences in migration policies, healthcare system organisation, and socioeconomic circumstances.

Our results have implications for clinical work and research. First, these findings raise intriguing questions regarding preparedness and resilience of healthcare systems in highly anticipated burden of long COVID. This implies that the healthcare system that dealt with the acute consequences of COVID-19 now also has to consider a new situation of long-term and indirect consequences of the pandemic, which puts new demands on healthcare professionals and society. Second, the higher risk of long COVID in ethnic minorities is a major concern for equity in health, and addressing this health problem may require multisectoral response, funding, care, and treatment approaches, which are culturally acceptable to these populations. To decrease health inequalities, we must address a variety of public policy issues, combining initiatives that target economic and social injustices with targeted attention to marginalized communities and groups such as ethnic minorities. This could entail using legislation, taxation, regulation, and policy to ensure a more equitable distribution of wealth, power, and income. To address the impact of long COVID among ethnic minorities, a concerted effort is needed, particularly focusing on equal access to high-quality housing and improved working conditions for everyone, advocacy activities for COVID-19 vaccines, and adoption or continuation of preventive measures such as avoiding close contact with confirmed cases of COVID-19, avoiding poorly ventilated areas, social distancing, use of face masks in crowded environment as well as handwashing and use of sanitizers. Lastly, future research should also focus on understanding key drivers of long COVID and the impact of long COVID on sick leave and labour market participation among ethnic minorities.

## Supporting information

**S1 Table. List of International Classification of Diseases (ICD-10) codes included.** COVID-19, Coronavirus Disease 2019.
(DOCX)

**S2 Table. Individuals who had first time tested positive for SARS-CoV-2 between January 2020 and August 2022 by largest countries of origin.** Data are in median (IQR) or n (%). *Family income was the total household disposable income among patients with COVID-19 in the specific calendar year. $^{\S}$CCI composed myocardial infarction, congestive heart failure, peripheral vascular disease, cerebrovascular disease, chronic obstructive pulmonary disease, rheumatic disease, dementia, peptic ulcer disease, hemiplegia, diabetes without complications, diabetes with complications, mild liver disease, moderate to severe liver disease, renal disease, malignancy, metastatic cancer, and AIDS. AIDS, acquired immunodeficiency syndrome; CCI, Charlson comorbidity index; COVID-19, Coronavirus Disease 2019; IQR, interquartile range; NA, not applicable; SARS-CoV-2, Severe Acute Respiratory Syndrome Coronavirus 2.
(DOCX)

**S3 Table. Age-standardised incidence rates of long COVID per 100,000 person-years by region of origin.** Northern Europe indicates Northern Europe other than Denmark. *Standardised to 2020 Danish population age distribution. CI, confidence interval; IR, incidence rate.
(DOCX)

**S4 Table. Hazard ratios of long COVID diagnosis among individuals aged 18–60 years and >60 years by region of origin.** Northern Europe indicates Northern Europe other than Denmark. The adjusted model composed age, sex, civil status, education, family income, and CCI. CCI, Charlson comorbidity index; CI, confidence interval; HR, hazard ratio.
(DOCX)

**S5 Table. Hazard ratios of long COVID diagnosis by age group.** Northern Europe indicates Northern Europe other than Denmark. The adjusted model composed age, sex, civil status, education, family income, and CCI. CCI, Charlson comorbidity index; CI, confidence interval; HR, hazard ratio.
(DOCX)

**S6 Table. Hazard ratios of long COVID diagnosis by sex.** Northern Europe indicates Northern Europe other than Denmark. The adjusted model composed age, sex, civil status, education, family income, and CCI. CCI, Charlson comorbidity index; CI, confidence interval; HR, hazard ratio.
(DOCX)

**S7 Table. Hazard ratios of long COVID diagnosis for males and females by region of origin.** Northern Europe indicates Northern Europe other than Denmark. The adjusted model composed age, sex, civil status, education, family income, and CCI. CCI, Charlson comorbidity index; CI, confidence interval; HR, hazard ratio.
(DOCX)

**S8 Table. Hazard ratios of long COVID diagnosis among hospitalised and nonhospitalised individuals by region of origin.** Northern Europe indicates Northern Europe other than Denmark. The adjusted model composed age, sex, civil status, education, family income, and CCI. CCI, Charlson comorbidity index; CI, confidence interval; HR, hazard ratio.
(DOCX)

**S9 Table. Hazard ratios of long COVID diagnosis by number of doses of COVID-19 vaccine.** Northern Europe indicates Northern Europe other than Denmark. *Estimates are not displayed due to small numbers in accordance with Danish Data Protection Act. The adjusted model composed age, sex, civil status, education, family income, and CCI. CCI, Charlson comorbidity index; CI, confidence interval; HR, hazard ratio.
(DOCX)

**S10 Table. Odds ratio of hospital contacts related to specific symptoms 6 months after COVID-19 diagnosis compared with 6 months before COVID-19 diagnosis by region of origin.** Northern Europe indicates Northern Europe other than Denmark. Hospital contacts related to cardiopulmonary symptoms included dyspnoea (difficulty in breathing), cough, and chest pain as a composite outcome. Hospital contacts related to any long COVID symptoms included fatigue, headache, dyspnoea (difficulty in breathing), cough, chest pain, depression, and/or anxiety as a composite outcome. The adjusted model composed age, sex, civil status, education, family income, and CCI. CCI, Charlson comorbidity index; CI, confidence interval; COVID-19, Coronavirus Disease 2019; OR, odds ratio.
(DOCX)

**S11 Table. Odds ratio of hospital contacts related to any long COVID symptoms 6 months after COVID-19 diagnosis compared with 6 months before COVID-19 diagnosis by largest countries of origin.** Hospital contacts related to any long COVID symptoms included fatigue, headache, dyspnoea (difficulty in breathing), cough, chest pain, depression and/or anxiety as a composite outcome. The adjusted model composed age, sex, civil status, education, family income, and CCI. CCI, Charlson comorbidity index; CI, confidence interval; COVID-19, Coronavirus Disease 2019; OR, odds ratio.
(DOCX)

**S12 Table. Odds ratios of hospital contacts related to any long COVID symptoms by largest countries of origin.** Hospital contacts related to any long COVID symptoms included fatigue, headache, dyspnoea (difficulty in breathing), cough, chest pain, depression, and/or anxiety as a composite outcome. The adjusted model composed age, sex, civil status, education, family income, and CCI. CCI, Charlson comorbidity index; CI, confidence interval; OR, odds ratio.
(DOCX)

**S13 Table. Pattern of general practitioner contacts before and during COVID-19 pandemic among individuals diagnosed with COVID-19.** Northern Europe indicates Northern Europe other than Denmark.
(DOCX)

**S1 Fig. Directed acyclic graphs for confounders assessment for the association between migration and long COVID diagnosis.** Green lines indicate the pathway of mediators. Black lines indicate the pathway of confounders. Blue circles indicate mediating factors. White circles indicate confounding factors. Age, sex, civil status, comorbidities, education, and income were identified as confounders.
(TIFF)

**S2 Fig. Pattern of hospital contacts related to specific symptoms in pre-COVID and during COVID periods.** Northern Europe indicates Northern Europe other than Denmark. Hospital contacts related to cardiopulmonary symptoms included dyspnoea (difficulty in breathing), cough, and chest pain as a composite outcome. Hospital contacts related to any long COVID symptoms included fatigue, headache, dyspnoea (difficulty in breathing), cough, chest pain, depression, and/or anxiety as a composite outcome.
(TIFF)

**S1 Appendix. Data used for calculation of age-standardised incidence rate of long COVID diagnosis.**
(XLSX)

**S1 Study Protocol. COVID-19 long-term outcomes/heath consequences among migrants in Denmark: A nationwide register-based study.**
(DOCX)

**S1 RECORD Checklist. The RECORD statement—Checklist of items, extended from the STROBE statement, that should be reported in observational studies using routinely collected health data.**
(DOCX)

## Author Contributions

**Conceptualization:** George Frederick Mkoma, Charles Agyemang, Thomas Benfield, Mikael Rostila, Agneta Cederström, Jørgen Holm Petersen, Marie Norredam.

**Data curation:** George Frederick Mkoma, Marie Norredam.

**Formal analysis:** George Frederick Mkoma, Jørgen Holm Petersen.

**Funding acquisition:** Charles Agyemang, Thomas Benfield, Mikael Rostila, Marie Norredam.

**Investigation:** George Frederick Mkoma, Marie Norredam.

**Methodology:** George Frederick Mkoma, Charles Agyemang, Thomas Benfield, Mikael Rostila, Agneta Cederström, Jørgen Holm Petersen, Marie Norredam.

**Project administration:** George Frederick Mkoma, Marie Norredam.

**Resources:** George Frederick Mkoma, Marie Norredam.

**Software:** George Frederick Mkoma, Agneta Cederström.

**Supervision:** Charles Agyemang, Thomas Benfield, Mikael Rostila, Jørgen Holm Petersen, Marie Norredam.

**Validation:** George Frederick Mkoma, Marie Norredam.

**Visualization:** George Frederick Mkoma.

**Writing – original draft:** George Frederick Mkoma.

**Writing – review & editing:** George Frederick Mkoma, Charles Agyemang, Thomas Benfield, Mikael Rostila, Agneta Cederström, Jørgen Holm Petersen, Marie Norredam.

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
