## [Editor Report · Decision Letter 0]

21 Aug 2023

Dear Dr Mkoma, 

Thank you for submitting your manuscript entitled "Risk of long COVID and associated symptoms after acute SARS-COV-2 infection in ethnic minorities: a Danish nationwide cohort study" for consideration by PLOS Medicine.

Your manuscript has now been evaluated by the PLOS Medicine editorial staff and I am writing to let you know that we would like to send your submission out for external peer review.

Please re-submit your manuscript within two working days, i.e. by Aug 23 2023 11:59PM.

Kind regards,

Alexandra Schaefer, PhD

Associate Editor

PLOS Medicine

---

## [Decision Letter · Decision Letter 1]

6 Oct 2023

Dear Dr. Mkoma,

Thank you very much for submitting your manuscript "Risk of long COVID and associated symptoms after acute SARS-COV-2 infection in ethnic minorities: a Danish nationwide cohort study" (PMEDICINE-D-23-02370R1) for consideration at PLOS Medicine. 

Your paper was evaluated by an associate editor and discussed among all the editors here. It was also discussed with an academic editor with relevant expertise, and sent to three independent reviewers, including a statistical reviewer. The reviews are appended at the bottom of this email and any accompanying reviewer attachments can be seen via the link below:

[LINK]

In light of these reviews, I am afraid that we will not be able to accept the manuscript for publication in the journal in its current form, but we would like to consider a revised version that addresses the reviewers' and editors' comments. Obviously we cannot make any decision about publication until we have seen the revised manuscript and your response, and we plan to seek re-review by one or more of the reviewers. 

We expect to receive your revised manuscript by Oct 27 2023 11:59PM. Please email me (aschaefer@plos.org) if you have any questions or concerns.

We look forward to receiving your revised manuscript. 

Sincerely,

Alexandra Schaefer, PhD

Associate Editor

PLOS Medicine

plosmedicine.org

GENERAL COMMENTS

Please respond to all editor and reviewer comments.

1) Please include page numbers and line numbers in the manuscript file. Use continuous line numbers (do not restart the numbering on each page).

2) Please cite your Supporting Information as outlined here: https://journals.plos.org/plosmedicine/s/supporting-information

3) Please remove the Ethics statement, the Financial Disclosure statement, Data availability statement and Competing Interest statement from the main body of the manuscript. These details should only be provided in the corresponding section in the manuscript submission form.

4) When you refer to “Northern Europe”, please ensure that you mean “Northern Europe other than Denmark”.

ACADEMIC EDITOR COMMENTS

This is an interesting paper that uses a very robust dataset and well-thought-out analyses. My main feedback would be to orient discussion to talk about the causal underpinnings of these disparities and how they could be driven by systemic inequities. In particular, it seems quite important to emphasize these points given the robust existing literature surrounding them compared to biologic differences and reasons for these disparities.

1) Abstract

*Should read “less is known about long COVID in these populations”

*First line under Methods and Findings is not a complete sentence

2) Introduction

*The introduction highlights several studies that have identified disparities across multiple studies but would benefit from clear discussion also highlighting the systemic inequities in access and treatment that have likely led to these disparities in outcomes. In this discussion it is important to recognize that even in settings like Denmark with a robust public health infrastructure and free testing and vaccination, systemic inequities still likely exist that lead to different accesses to trusted information, knowledge about resources, time away from work, etc (could see Saarinen et al. in PLOS Medicine for an example). Any of these that could lead to differences in healthcare utilization that impact outcomes.

3) Methods

*Is there data on disparities in testing and vaccination uptake? Just because it is free for everyone doesn’t mean access is the same as many other factors going into access (see Levesque et al. for a framework on components of access to health care).

*I am not sure you can say that primary outcome definition is “complications persisting beyond the acute COVID-19 infection that cannot be explained by alternative diagnosis”. This would likely need an adjudication process and detailed review of health records (or at least documentation of a standardized set of tests excluding other reasons). This is conceptually what the authors are trying to capture, but a more appropriate definition is documented ICD-10 codes for long COVID and ICD-10 codes for related symptomatology.

*Would mediation analyses examining impact of hospitalization and/or vaccination be possible?

4) Results

*I wonder if there is the possibility for ascertainment bias particularly related to hospital contacts. If hypotheses about inequities in access are true, ascertainment of these symptoms may have been lower among ethnic minorities compared to native Danes. If ethnic minorities then increased their health care utilization after a COVID-19 diagnose, this would then appear to be an increase in these symptoms when it is really driven by differences in health care seeking behavior. Are there differences in baseline healthcare utilization (not necessarily symptoms but number of primary care visits or other vaccine uptake, etc.)? This is an issue that plagues most research on long COVID, and it is challenging to feel confident in findings in baseline.

*Figure 4 seems to somewhat suggest that patterns of disparities existed prior to COVID, but they prevalence of symptoms increased by 3 times. For example, the shape of the differences across ethnic groups remains the same. This would somewhat go against the conclusions, and I think highlights some risk in overinterpreting the findings and occasional statistically significant values.

*What were differences in symptomology between ethnic minorities when only restricted to pre-COVID period? I think differences pre-COVID would again suggest other underlying etiologies for disparities (e.g., systemic inequities).

5) Discussion

*The data is excellent, and the methods are solid, but I think interpretation of what these findings mean is lacking. The discussion is focused primarily on biomedical considerations, but none of these truly explain why ethnic minorities would have increased risk of long COVID. Ethnic minorities were younger, and age was also adjusted for. There was increased hospitalizations and use of ICU, but why? Is there a biologic reason or is the differences in the complex socio-economic, -cultural, - political milieu that they live in that create systemic inequities and ultimately disparities in outcomes. I do not think it is biological and I think onus would be to prove that there are given existing robust evidence on systemic inequities that already present a plausible causal pathway. All of the variable differences discussed potentially have different causal underpinnings.

*I would highly consider consulting the frameworks and thinking of authors like Camara Jones and Rhea Boyd to gain additional insights for how to conceptualize design and causal considerations when considering disparities (https://academic.oup.com/aje/article/154/4/299/61900). It mostly applies to racism in the United States by a lot of the principles are applicable. The authors already apply thoughtful use of DAGs and I think this could flesh them out more and allow at least discussion of the main causal pathways including why migration may lead to increased hospitalizations.

FINANCIAL DISCLOSURE

The funding statement should include: specific grant numbers, initials of authors who received each award, URLs to sponsors’ websites. Also, please state whether any sponsors or funders (other than the named authors) played any role in study design, data collection and analysis, the decision to publish, or preparation of the manuscript. If they had no role in the research, include this sentence: “The funders had no role in study design, data collection and analysis, decision to publish, or preparation of the manuscript.”

ABSTRACT

1) Abstract Methods and Findings:

*Please revise the first sentence

*Please provide brief demographic details of the study population (e.g. sex, age).

*In the last sentence of the Abstract Methods and Findings section, please describe the main limitation(s) of the study's methodology.

2) Abstract Conclusions:

*Please change ‘in this population’ to ‘in these populations’

3) Please ensure that all numbers presented in the abstract are present and identical to numbers presented in the main manuscript text.

4) PLOS Medicine requests that main results are quantified with 95% CIs as well as p values. When reporting p values please report as p<0.001 and where higher as the exact p value p=0.002, for example. For the purposes of transparent data reporting, if not including the aforementioned please clearly state the reasons why not.

5) Please include any important dependent variables that are adjusted for in the analyses.

6) Throughout, suggest reporting statistical information as follows to improve clarity for the reader “22% (95% CI [13%,28%]; p</=)”. Please amend throughout the abstract and main manuscript.

7) Please note the use of commas to separate upper and lower bounds, as opposed to hyphens as these can be confused with reporting of negative values.

8) When a p value is given, please specify the statistical test used to determine it.

AUTHOR SUMMARY

At this stage, we ask you to provide a short, non-technical Author Summary of your research to make findings accessible to a wide audience that includes both scientists and non-scientists. The Author Summary should immediately follow the Abstract in your revised manuscript. This text is subject to editorial change and should be distinct from the scientific abstract. Please see our author guidelines for more information: https://journals.plos.org/plosmedicine/s/revising-your-manuscript#loc-author-summary.

The summary should include 2-3 single sentence, individual bullet points under each of the questions. The last bullet point should describe the main limitation(s) of the study's methodology.

It may be helpful to review currently published articles for examples which can be found on our website here https://journals.plos.org/plosmedicine/

INTRODUCTION

1) If possible, include the citations for the relevant studies in this sentence: “Overall, the previous studies have several shortcomings, including studies were based on a single hospital setting or localized area, the studies did not compare symptoms distribution before and after COVID-19 diagnosis, and most of the studies were survey-based.”

METHODS AND RESULTS 

1) For all observational studies, in the manuscript text, please indicate: (1) the specific hypotheses you intended to test, (2) the analytical methods by which you planned to test them, (3) the analyses you actually performed, and (4) when reported analyses differ from those that were planned, transparent explanations for differences that affect the reliability of the study's results. If a reported analysis was performed based on an interesting but unanticipated pattern in the data, please be clear that the analysis was data-driven.

2) Did your study have a prospective protocol or analysis plan? Please state this (either way) early in the Methods section.

3) PLOS Medicine requests that main results are quantified with 95% CIs as well as p values. When reporting p values please report as p<0.001 and where higher as the exact p value p=0.002, for example. For the purposes of transparent data reporting, if not including the aforementioned please clearly state the reasons why not.

4) Please include any important dependent variables that are adjusted for in the analyses.

5) Suggest reporting statistical information as detailed above – see under ABSTRACT

6) Please present numerators and denominators for percentages, at least in the Tables [not necessarily each time they're mentioned].

7) Please ensure that the study is reported according to the RECORD guideline and include the completed RECORD checklist as Supporting Information. When completing the checklist, please use section and paragraph numbers, rather than page numbers. Please add the following statement, or similar, to the Methods: "This study is reported as per the Reporting of studies Conducted using Observational Routinely-collected health Data (RECORD) statement (S1 Checklist)."

8) Please define abbreviations at first use (e.g., PCR).

9) When referring to ethnicities like 'North African' as a group, please use the plural, e.g., 'North Africans'. Please revise throughout the entire manuscript.

10) “Overall, 6479 native Danes and 755 ethnic minorities died within 6 months after COVID-19 diagnosis.” – please add percentages.

DISCUSSION

1) Please present and organize the Discussion as follows: a short, clear summary of the article's findings; what the study adds to existing research and where and why the results may differ from previous research; strengths and limitations of the study; implications and next steps for research, clinical practice, and/or public policy; one-paragraph conclusion. Please remove any subheadings.

2) The discussion in its current form does not seem to be of sufficient depth - please revise and take into account the comments made by the academic editor and reviewers.

TABLES

1) Please define all abbreviations used in the tables (including those in Supporting Information files).

2) Table 1: Please define ‘IQR’. Please add details about how ‘Family income’ was defined.

3) Table 3: Please define the asterisk and ‘§’.

FIGURES

1) For all Figures, please ensure that you have complied with our figures requirements http://journals.plos.org/plosmedicine/s/figures.

2) Please define all abbreviations used in figures (figure, figure legend, figure description, including those in Supporting Information files).

3) Please consider avoiding the use of red and green in order to make your figure more accessible to those with color blindness (including those in Supporting Information files).

4) Please provide titles and legends for all figures (including those in Supporting Information files).

5) Figure 1: Typically, a flowchart goes from top to bottom and the figure seems to be missing important details, e.g., the definition of the ethnic minorities or the cause of death, - please revise and add sufficient details.

6) Figure 4: Please see the comment made by Reviewer #2. If you decide to keep the current format, please indicate in the figure caption the meaning of the bars and whiskers.

SUPPLEMENTARY MATERIAL

1) Please define all abbreviations used in figures and tables (e.g., ‘ICD’).

3) Figure S1: Please describe the meaning of the colors of the lines and dots.

4) Table S2: Please see comments for Table 1

5) Table S3: Please change to “Age groups (years)”.

6) Table S6: The asterisk is missing in the description.

REFERENCES

1) PLOS uses the numbered citation (citation-sequence) method and first six authors, et al.

2) Please ensure that journal name abbreviations match those found in the National Center for Biotechnology Information (NCBI) databases (http://www.ncbi.nlm.nih.gov/nlmcatalog/journals) and are appropriately formatted and capitalized.

3) Where website addresses are cited, please specify the date of access.

4) Please also see https://journals.plos.org/plosmedicine/s/submission-guidelines#loc-references for further details on reference formatting. 

Comments from the reviewers:

Reviewer #1: This review looks at the statistical elements of the paper by Mkoma et al, a national retrospective cohort study of people diagnosed with CIVOD-19 in Denmark, investigating ethnic differences in subsequent long COVID symptoms.

In general, the methods used are good, with Cox PH models for time to long COVID diagnosis, and logistic regression for events within fixed time periods relative to positive testing. I do, however, have a few observations.

The study population is defined by the first time someone tested positive, but I wonder what the chances are that someone first testing positive towards the end of the study period had truly never had COVID before? Does this matter to the analysis at all? Also, calendar time in general does not appear to have been taken into account in the analysis in any way. Assuming the different variants have different propensity to lead to long-COVID, would there be any value in looking at the different waves of infections in this analysis? Infection wave, or predominant variant, might come under the group of variables on the causal pathway, and not adjusted for in most analyses, but could be a predictor in secondary analyses.

There is a sentence at the end of the "Outcomes" section on page 4: "The analysis was restricted to the population experiencing these groups of symptoms within 6 months before COVID-19 diagnosis, 0 to 4 weeks (acute phase of COVID-19 infection), and >4 weeks to 6 months after COVID-19 diagnosis." This confused me - does it mean that people who never experienced these symptoms were excluded from this analysis, or does it mean that this analysis only considers outcomes that occurred within these time windows?

In the "Covariates" section, it states that age was initially considered as a continuous variable, and then categorised. Was there a particular reason for the categorisation used?

Subgroup analyses are done by ethnic minority groups and by age, but not by sex. Would that not be worth looking at? Also, when looking at subgroups, it is usually best to use interaction terms within regression models to test for differences in associations. This might help to prevent reading too much into the subgroup analyses, for example, on page 8, the sentence "Further analysis showed that individuals not receiving COVID-19 vaccine exhibited greater risk of long COVID diagnosis than individuals vaccinated; and the association was found in native Danes only (aHR 1.47; 95% CI 1.33-1.63)" - this gives the impression that this association is limited to native Danes, but it could easily be the case that the same association exists in other ethnic groups, but fails to reach statistical significance due to smaller sample size. However, a test of interaction between ethnic group and vaccination status might indicate whether the association is actually different between ethnic groups.

In the tables, there is information about the denominators, or the numbers of people testing positive for COVID for the first time, in relation to various factors (Table 1). The tables then move straight on to reporting hazard ratios. I did not see any raw data showing the incidence of long COVID. Could some information be given about the numbers of diagnoses and person-years of follow-up, and event rates, within subgroups of interest? Perhaps something for the supplementary tables.

Reviewer #2: Thank you for the opportunity to review this study on the association between ethnicity and long COVID (diagnoses and selected symptoms) performed using Danish registries. Overall the study has been well planned and analysed and there is a thorough discussion.

There are a number of points I would recommend are addressed prior to publication. Most of them are minor, with a number of more substantial comments related to additional analyses that should be performed. Substantial then minor comments in the order that they appear in the manuscript are as follows:

Substantial comments

1. Results - Risk of long COVID diagnosis: the main results show that North African, Middle Eastern, Eastern European and Asian ethnicities all had higher hazards of long COVID diagnosis, and as noted these are all groups who were more likely to be hospitalised. Is it possible that the effects seen by ethnicity are actually just effects due to hospitalisation? See comments below for how to investigate this.

2. Results - Risk of long COVID diagnosis: in order to try and address the point above I think the authors have performed the analysis included in Table 2, but this only looks at the effect of hospitalisation within each group of ethnicity, it doesn't show whether the overall ethnicity effect observed in the main analysis is due to hospitalisation. I would recommend an additional set of analyses where the main analysis (i.e. figure 2) is repeated within a restricted cohort of people - those who were not hospitalised (i.e. to see if there is still an effect of ethnicity when the effect of hospitalisation is removed). It would also be worth performing this analysis only in the group who were hospitalised.

3. Discussion: the discussion should be updated based upon the results of 2.

4. Discussion: a related point that isn't touched on in the discussion but it would be good to see a comment on is what about the fact that those who were hospitalised are more likely to get a long COVID diagnosis if the clinician knew they were hospitalised, and more ethnic minorities were hospitalised? I am not sure there is much that can be done about this bias but it should be discussed, in conjunction with the new results from 3. above.

5. Overall thought on substantial comments: it is still important to show and highlight that ethnic disparities continue into long COVID as the authors have done, but there is an opportunity here to tease out the role of prior hospitalisation which has been missed.

Minor comments

1. Abstract - Background: "infection rates and hospitalisation" - also death from COVID e.g. see e.g. see Mathur et al https://doi.org/10.1016/S0140-6736(21)00634-6 

2. Abstract - Methods and Findings: "in both unadjusted and adjusted models" - I would remove this as its already specified that the results discussed are adjusted

3. Abstract - Methods and Findings: "greater" - should this be "greatest"?

4. Introduction: (same comment as abstract): "infection rates, hospitalisation and severe morbidity" - more deaths also reported previously in ethnic minority groups

5. Introduction: "country origin" - I guess this should be "country of origin"

6. Introduction: "taking into account….hospitalisation and vaccination status" - I am not sure that the study takes account of either of these currently (see later comments).

7. Methods - Outcome: "fatigue, headache, cardiopulmonary symptoms" - it would be good to see reasoning/references for why these three types were selected for analysis.

8. Methods - Outcome: "The analysis was restricted to the population experiencing these groups of symptoms" - this wording is quite confusing. The study population has already been defined previously as "all individuals residing in Denmark who had first-time tested positive for SARS-COV-2 (COVID-19 diagnosis) aged 18 years or older from January 1, 2020 to August 31, 2022.". Here is sounds like there is further restriction on the study population but I think what is meant is that these are the outcomes that were studied, please rephrase.

9. Methods - Statistical analysis: "and the risk of long COVID diagnosis" - I think this should be "hazard of long COVID diagnosis"?

10. Methods - Statistical analysis: "covariates like in the Cox models" - "covariates as the Cox models"?

11. Methods - Statistical analysis: last 2 sentences beginning "All hazard ratios.." - I wasn't sure how this relates to what is presented in Figure 4 (see also specific comment on Figure 4).

12. Results - hospital contacts related to long COVID symptoms: The probabilities figure (figure 4) is unclear, I would recommend replacing it with a forest-plot type figure (like Figure 2) containing the results (ORs) in Table S4.

Reviewer #3: This is a generally well-presented study examining the association between the risk of long covid and ethnic origin in Denmark. The authors found that belonging to an ethnic minority group was significantly associated with an increased risk of long COVID compared to native Danes. These are important findings and very relevant to the detection, care and support for people with long COVID and for efforts to reduce health inequalities in the after effects of the pandemic.

Below are some comments that I hope the authors would find helpful in improving the clarity of their manuscript. These are categorised by sections and subheadings. It would be good if the authors include line numbers on every page if they resubmit a revised manuscript.

Introduction

Final paragraph - Can you clarify what 'cardiopulmonary symptoms'? 

Methods

Setting - can you clarify the duration of free access to testing? 

Data sources and study population - can you clarify what is meant by 'symptoms related to hospital contacts'?

Were the symptoms only recorded in hospitalised patients or were primary care data also included? 

Region and country of origin - you state that "Individuals originating outside Denmark and their descendants formed the ethnic minority population. Participants originating and/or born in Denmark (native Danes) constituted the reference group". It is not clear from this statement which group were second generation ethnic minorities born in Denmark classified as. Please clarify.

Outcome - in the definition of the Long Covid diagnosis, 'complications' need to be defined and listed too. 

By 'hospital' contacts do you mean hospital admissions, casualty attendance, outpatient attendance or all? This needs classification.

It is not clear weather 'anxiety' and 'depression' as 'secondary outcomes' were examined as separate diagnosis or symptoms of long covid. 

Covariates - The 'length of residency' variable. was it equivalent to age for the control group? please clarify.

Results

Participants characteristics - in the sentence 'overall 6479 native Danes and 755 ethnic minorities died within six months' please modify to 'people from ethnic minorities'.

Risk of long COVID diagnosis - it would be helpful to include absolute numbers besides the hazard ratio and the confidence interval particularly when looking at risk by country of origin to give the reader an idea about the sample size for each category. I can see it's included in figure 3 but it would be good to include in the text too.

In the second paragraph of this section it is not entirely clear what the hazard ratio for those from sub-Saharan Africa aged more than 60 years is in comparison to. This needs clarification in the text.

In the DAG (Figure S1), Why is the outcome healthcare contact rather than long COVID diagnosis?

Was there a difference in risk between 'immigrants' and 'descendants' as classified in table 1? did you conduct any subgroup analysis to look at that? it would be helpful to see that if we were going to hypothesise about the causes of the increased risk and ethnic minorities. 

in table S2 some of the variables had a category of 'missing' while others did not. Does that mean those with no missing category did not have any missing observations at all? 

From Table S2, it seems some of the most striking differences in vaccination status were for three doses. did you conduct analysis by the number of doses of vaccination?

it is not clear to me how the analysis was conducted for the results in table S3. Was age entered in the model as an independent variable? Did you conduct two subgroup models for the different age groups? please clarify. 

Figure 2 - state 'northern Europe other than Denmark' for this category. 

More information about the analysis informing the results in Figure 4 are needed. What is the comparison group here?

More information is needed about the analysis comparing the period before COVID diagnosis and after in terms of symptoms. When comparing the symptoms after COVID diagnosis did you adjust for baseline symptoms before COVID-19 diagnosis when comparing between the ethnic group categories? Some of the ethnic group categories had higher baseline risk of the symptoms.

Discussion

You discuss potential explanations for the observed increased risk of long COVID in ethnic minority groups. However some of these could have been adjusted for in the model as mediators to see if they partially explain the observed significant associations. 

I think the discussion around hospitalisation and age as risk factors confuses the narrative of the paper a little bit. These factors are established factors in the severity of covid, and also there is emerging evidence of a link with long COVID. The important thing in this analysis is that these factors are accounted for when testing the relationship of ethnicity with long covid rather than confusing that narrative with multiple subgroup analysis.

in terms of the discussion around intensive care being a confounder in the relationship why was this not examined? Was this data not available?

Again the discussion around comorbidities being a possible explanation of the increased risk in ethnic minority groups doesn't need to be speculative as these were included in the models.

The authors then suggest biological variation as a possible explanation after adjusting for comorbidities. For example they mention 'immunological factors' and 'circulatory system'. However the most likely explanation is that there is residual confounding in terms of socioeconomic disadvantage. This needs discussion. This includes a discussion around the completeness, the quality and validity of the data that's assessing socio economic disadvantage in this study. 

The evidence around the prevalence of long COVID in ethnic minorities is not necessarily consistent. For example in the UK the data from the office for National Statistics does not show higher prevalence of long COVID among ethnic minorities and lower prevalence in some of these groups. The discussion can include potential reasons for the inconsistency of the evidence.

Also there needs to be a discussion around the outcome assessment and the context for that. How is long covid diagnosis reached in Denmark? Could healthcare utilisation patterns and behaviours partially influence who gets the diagnosis?

in the discussion you mention the contrast of clinical based diagnosis versus symptom based diagnosis. It is not very clear what this means. The long COVID diagnosis should be primarily based on symptoms as there is no biomarker or diagnostic imaging for it at the moment. 

In your discussion of the implications of these findings, please mention aspects of prevention as a means of reducing health inequalities.

[LINK]

---

## [Decision Letter · Decision Letter 2]

1 Dec 2023

Dear Dr. Mkoma,

Thank you very much for re-submitting your manuscript "Risk of long COVID and associated symptoms after acute SARS-COV-2 infection in ethnic minorities: a nationwide register-linked cohort study in Denmark" (PMEDICINE-D-23-02370R2) for review by PLOS Medicine.

Thank you for your detailed response to the editors' and reviewers' comments. I have discussed the paper with my colleagues and the academic editor, and it has also been seen again by all three original reviewers. The changes made to the paper were mostly satisfactory to the reviewers. As such, we intend to accept the paper for publication, pending your attention to the editorial comments below in a further revision. When submitting your revised paper, please include a detailed point-by-point response to the editorial comments.

[LINK]

We expect to receive your revised manuscript within 1 week. Please email me (aschaefer@plos.org) if you have any questions or concerns.

If you have any questions in the meantime, please contact me (aschaefer@plos.org) or the journal staff on plosmedicine@plos.org.  

We look forward to receiving the revised manuscript by Dec 08 2023 11:59PM.   

Sincerely,

Alexandra Schaefer, PhD

Associate Editor 

PLOS Medicine

plosmedicine.org

Requests from Editors:

ACADEMIC EDITOR COMMENTS

1) Clarity on the outcome is needed: I think they need to be clear that that the outcome was an ICD-10 diagnosis of long COVID. Per the ICD-10 definitions, this is supposed to mean complications persisting beyond acute illness. But it is key to recognize that just because a clinician records an ICD-10 diagnosis, it doesn't mean that they did their due diligence. For clarity, I would write that the outcome was ICD-10 diagnosis of long COVID, and then say this is supposed to indicate a scenario where complications with no other explanations exist. Discussing it in this manner at least acknowledges that there may be a discrepancy.

2) I also still think that Fig S2 strongly suggests that there may be an underlying etiology of disparities in symptomatology (and therefore long COVID diagnosis) that is unrelated to any biologic reasons. I appreciate that in Table 4 the magnitude of some but certainly not all the odds ratios are greater in ethnic minorities. This could easily be by chance and there is no formal statistical assessment to see if disparities were indeed greater in magnitude during the COVID time periods. In several analyses, the point estimate is greater, but confidence intervals overlap. The authors do discuss this in their discussion, but I still think there is an overinterpretation of these findings. I think these point needs to be clearer. Although it is possible that these findings indicate an increase in long COVID, it may also be entirely true that there are no real differences (perhaps only differences in reporting and recorded diagnoses). Even if differences are real, they are likely more subtle than most HRs would suggest. For example, taking a difference-in-difference type approach to the Table 4 results would show perhaps only small increases in the disparities when taking into account the disparities that already existed. I know this goes against the author's interpretation but I think needs to be taken into account.

EDITORIAL COMMENTS

1) Methods/Limitations: Please discuss in more detail whether you considered a single long COVID-related symptom as “Long COVID”. Please discuss whether there is a possibility that symptoms might be unrelated to COVID/long COVID. In this regard, we also feel it is important that you emphasize the low numbers in some groups (e.g., consultations for individual symptoms; table 4) in the limitations.

2) Limitations: Please discuss the influence on medications/treatment on your study design/results and mention whether these data were available in the registries and, if yes, why you chose not to incorporate these data into this study.

GENERAL

1) Please carefully review the terms used to describe ethnic minorities in your study. For example, "Iraqis" does not seem appropriate because the groups are a combination of immigrants and their first-generation descendants. We suggest using, for example, "people of Iraqi origin" or, e.g., “people of North African origin" (as you have already done on occasion). Please make sure these terms are used consistently throughout the paper.

2) It seems that line numbering has only been added to the marked-up version of your manuscript. Please check.

ABSTRACT

l.46: Please define ‘CI‘ at first use.

AUTHOR SUMMARY

The Author summary in its current form is rather long, particularly ‘Why was this study done?’ and ‘What do these findings mean?’,. Please try to shorten the individual bullet point and to focus on core details. 

INTRODUCTION

l.130: The terms gender and sex are not interchangeable (as discussed in https://www.who.int/health-topics/gender); please be sure to use the appropriate term.

DISCUSSION

ll.518-519: „Living in these circumstances may sometimes be culturally influenced but can have

detrimental effect in individual’s recovery from COVID-19 infection.” – please provide a reference.

FIGURES

Figure 1: For easy comparison, we suggest adding percentages in the boxes.

TABLES

S12 Table: Please add a reference/footnote to S2 Table so that the reader can easily find the denominators used for the calculations in this table.

REFERENCES

Please use ‘accessed’ instead of ‘cited’ when specifying the date of access.

SOCIAL MEDIA

To help us extend the reach of your research, please provide any X (formerly known as Twitter) handle(s) that would be appropriate to tag, including your own, your coauthors’, your institution, funder, or lab. Please respond to this email with any handles you wish to be included when we tweet this paper.

Comments from Reviewers:

Reviewer #1: Alex McConnachie, Statistical Review

I thank the authors for their consideration of my original comments. All of their responses are good, and I have no further comments.

Reviewer #2: Thanks, all comments addressed well

Reviewer #3: 

1. Setting - can you clarify the duration of free access to testing? The authors responded: Testing has been free of charge throughout the COVID-19 pandemic and is still free of charge for all residents as it is financed by general taxes in Denmark (page 5 line 130-132). However, public testing was closed on April 1, 2023." - though I could only find a statement about access to care being free of charge on page 5. Could they add the specific response about testing to the text?

2. Region and country of origin - you state that "Individuals originating outside Denmark and their descendants formed the ethnic minority population. Participants originating and/or born in Denmark (native Danes) constituted the reference group". It is not clear from this statement which group were second generation ethnic minorities born in Denmark classified as. Please clarify. The authors responded stating the definition of descendants as"those who were born in Denmark from parents with foreign citizenship" but my point was that these overlap with the reference group of born in Denmark" - please clarify. 

3. Outcome - in the definition of the Long Covid diagnosis, 'complications' need to be defined and listed too. This has also not been adequately addressed by the authors. Is the correct word here "symptoms" rather than "complications"?

4. You discuss potential explanations for the observed increased risk of long COVID in ethnic minority groups. However some of these could have been adjusted for in the model as mediators to see if they partially explain the observed significant associations -The authors state that they did not adjust for medication but did not offer a justification for this. DAGs don't prevent the adjustment for mediation for the direct effect of the exposure on the outcome. 

5. In your discussion of the implications of these findings, please mention aspects of prevention as a means of reducing health inequalities - the authors have added a statement about avoiding close contact with cases, handwashing and use of sanitizers, though the last two are not the main preventive measures for covid given its airborne transmission: https://www.who.int/news-room/questions-and-answers/item/coronavirus-disease-covid-19-how-is-it-transmitted

[LINK]

---

## [Decision Letter · Decision Letter 3]

10 Jan 2024

Dear Dr. Mkoma,

Thank you very much for re-submitting your manuscript "Risk of long COVID and associated symptoms after acute SARS-COV-2 infection in ethnic minorities: a nationwide register-linked cohort study in Denmark" (PMEDICINE-D-23-02370R3) for review by PLOS Medicine.

I appreciate your detailed responses to the editors' and reviewers' comments. I have discussed the paper with my colleagues and the academic editor, and it has also been seen again by one of the original reviewers. The changes made to the paper were mostly satisfactory to the reviewer, but there seems to have been some misunderstanding about their comments regarding medication versus mediation due to a typo. Also, the reviewer feels that their comment about preventive measures has not been sufficiently addressed, with which the Editors agree. Therefore, we intend to accept the paper for publication, pending your attention to the editorial comments below in a further revision. When submitting your revised paper, please include a detailed point-by-point response to the editorial comments.

[LINK]

We expect to receive your revised manuscript within 1 week. Please email me (aschaefer@plos.org) if you have any questions or concerns.

We look forward to receiving the revised manuscript by Jan 17 2024 11:59PM.   

Sincerely,

Alexandra Schaefer, PhD

Associate Editor 

PLOS Medicine

plosmedicine.org

Requests from Editors:

1) Please revise your Author summary with an emphasis on using non-technical language. It is better to avoid words like " significantly " and please avoid reporting data/exact values ("1.2 to 1.4 times higher risk").

2) Under ‘What did the researchers do and find?’, please use non-technical language to describe your findings. Editorial suggestion: People of North African, Middle Eastern, Eastern European, and Asian origin were more likely to report cardiopulmonary symptoms (including dyspnea, cough, and chest pain) and any long COVID symptoms than native Danes, especially beyond 4 weeks to 6 months after COVID-19 diagnosis.

3) ll.592-595: Please provide reference (for the NICE guidelines; reference 5?)

4) Figure 1: For consistency, change "Danes" to "Native Danes" and add the definition of "Native Danes" in the figure description.

5) S2 Figure: We suggest that you start the axis of all four graphs at zero. If this is not possible, please show a break in the axis. Also, please include a note in the figure description stating that the y-axis is not identical for the four graphs. Please add a label for the x-axis, such as "time frame" or "time period".

6) S1 Appendix: This one has no reference in the main manuscript and does not have a title nor a description. Please revise.

Comments from Reviewers:

Reviewer #3: I thank the authors for responding to my previous comments.

1. Please include the statement in your response in the text of the paper to make this clear to international audiences: "As per Statistics Denmark definition, descendants of ethnic Danes and descendants of ethnic minorities are never classified into the same group. These two groups have different coding system based on the data from Statistics Denmark and can explicitly be separated from one another."

2. My comment from R2: "You discuss potential explanations for the observed increased risk of long COVID in ethnic minority groups. However some of these could have been adjusted for in the model as mediators to see if they partially explain the observed significant associations -The authors state that they did not adjust for medication but did not offer a justification for this. DAGs don't prevent the adjustment for mediation for the direct effect of the exposure on the outcome." - I am sorry there was a typo in my review. What I meant was 'mediation' not 'medication' - I was requested you justify why you haven't adjusted for mediation. Please do so.

3. In your discussion of the implications of these findings, please mention aspects of prevention as a means of reducing health inequalities - the authors have added a statement about avoiding close contact with cases, handwashing and use of sanitizers, though the last two are not the main preventive measures for covid given its airborne transmission: https://www.who.int/news-room/questions-and-answers/item/coronavirus-disease-covid-19-how-is-it-transmitted

I am afraid this comment has not been adequately addressed as the preventive methods listed stayed the same from the previous version however the authors acknowledge airborne transmission in their response. Please add preventive measures related to airborne transmission to the text.

[LINK]

---

## [Decision Letter · Decision Letter 4]

30 Jan 2024

Dear Dr Mkoma, 

On behalf of my colleagues and the Academic Editor, Aaloke Mody, I am pleased to inform you that we have agreed to publish your manuscript "Risk of long COVID and associated symptoms after acute SARS-COV-2 infection in ethnic minorities: a nationwide register-linked cohort study in Denmark" (PMEDICINE-D-23-02370R4) in PLOS Medicine.

I appreciate your thorough responses to the reviewers' and editors' comments throughout the editorial process. We look forward to publishing your manuscript, and editorially there is only one remaining minor point that should be addressed prior to publication. We will carefully check whether the change has been made. If you have any questions or concerns regarding these final requests, please feel free to contact me at aschaefer@plos.org.

Please see below the minor point that we request you respond to:

1) Figure 4: Please define 'CI' in the figure description.

PRESS

Sincerely, 

Alexandra Schaefer, PhD 

Associate Editor 

PLOS Medicine